# E-cadherin can limit the transforming properties of activating β-catenin mutations

David J Huels[1,†], Rachel A Ridgway[1,†], Sorina Radulescu[1], Marc Leushacke[2], Andrew D Campbell[1], Sujata Biswas[3,4], Simon Leedham[3,4], Stefano Serra[5], Runjan Chetty[5], Guenievre Moreaux[1], Lee Parry[6], James Matthews[6], Fei Song[7], Ann Hedley[1], Gabriela Kalna[1], Fatih Ceteci[1], Karen R Reed[6], Valerie S Meniel[6], Aoife Maguire[8], Brendan Doyle[1,8], Ola Söderberg[9], Nick Barker[2], Alastair Watson[10], Lionel Larue[11], Alan R Clarke[6] & Owen J Sansom[1,*]

## Abstract

Wnt pathway deregulation is a common characteristic of many cancers. Only colorectal cancer predominantly harbours mutations in *APC*, whereas other cancer types (hepatocellular carcinoma, solid pseudopapillary tumours of the pancreas) have activating mutations in β-catenin *(CTNNB1)*. We have compared the dynamics and the potency of β-catenin mutations *in vivo*. Within the murine small intestine (SI), an activating mutation of β-catenin took much longer to achieve Wnt deregulation and acquire a crypt-progenitor cell (CPC) phenotype than *Apc* or *Gsk3* loss. Within the colon, a single activating mutation of β-catenin was unable to drive Wnt deregulation or induce the CPC phenotype. This ability of β-catenin mutation to differentially transform the SI versus the colon correlated with higher expression of E-cadherin and a higher number of E-cadherin: β-catenin complexes at the membrane. Reduction in E-cadherin synergised with an activating mutation of β-catenin resulting in a rapid CPC phenotype within the SI and colon. Thus, there is a threshold of β-catenin that is required to drive transformation, and E-cadherin can act as a buffer to sequester mutated β-catenin.

**Keywords** APC; β-catenin; colorectal cancer; E-cadherin
**Subject Categories** Cancer; Cell Adhesion, Polarity & Cytoskeleton
**The EMBO Journal (2015) 34: 2321–2333**

See also: **GT Chen & ML Waterman** (September 2015)

## Introduction

Mutation of the *APC* (adenomatous polyposis coli) gene is the most common event in colorectal cancer (CRC). A recent study where a large number of CRCs (200) were sequenced showed mutation in over 70% of CRC (TCGA, 2012). It is proposed that the reason for *APC* mutation in CRC is due to its role as a negative regulator of the Wnt signalling pathway. APC is part of a large multiprotein destruction complex that targets β-catenin (CTNNB1, CATNB1) for degradation. In the absence of a Wnt ligand, APC in complex with AXIN, casein kinase 1 (CK1) and GSK3beta is required for the phosphorylation of β-catenin by GSK3, which targets β-catenin for degradation. Following a Wnt signal or *APC* mutation, this complex does not form and β-catenin is no longer targeted for degradation and accumulates in the nucleus. Within the nucleus, β-catenin interacts with T-cell factor-1/lymphoid-enhancing factor-1 (TCF/LEF1) transcription factors to drive transcription of TCF/LEF1 target genes (Kinzler & Vogelstein, 1996; Clevers, 2006).

CRC is unusual compared to other cancers in that loss of function mutations in *APC* are much more frequent than activating mutations in β-catenin (i.e. in exon 3), ~75% compared to ~5%, respectively (TCGA, 2012). In other cancers, such as hepatocellular cancer, activating mutations in the β-catenin gene are much more frequent, where the phosphorylation sites that target β-catenin for degradation are mutated (Cui *et al*, 2003; Edamoto *et al*, 2003; Ahn *et al*, 2014). As APC is a large protein, it has been hypothesised that functions other than just Wnt deregulation may be crucial for tumour

1   Cancer Research UK Beatson Institute, Glasgow, UK
2   A∗STAR Institute of Medical Biology, Singapore City, Singapore
3   Gastrointestinal Stem Cell Biology Laboratory, Wellcome Trust Centre for Human Genetics, University of Oxford, Oxford, UK
4   Translational Gastroenterology Unit, Experimental Medicine Division, Nuffield Department of Clinical Medicine, John Radcliffe Hospital, Oxford, Headington, UK
5   Department of Pathology, University Health Network/Toronto Medical Laboratories, Toronto, Canada
6   European Cancer Stem Cell Research Institute, Cardiff University, Cardiff, UK
7   Institute of Physiology, Justus-Liebig University Giessen, Giessen, Germany
8   Department of Histopathology, Trinity College Dublin, St James's Hospital, Dublin, Ireland
9   Department of Immunology, Genetics and Pathology Science for Life Laboratory, BMC, Uppsala University, Uppsala, Sweden
10  Norwich Medical School, University of East Anglia, Norwich, UK
11  Institut Curie, CNRS UMR3347, INSERM, U1021, Equipe labellisée – Ligue Nationale contre le Cancer, Orsay, France
    *Corresponding author. Tel: +44 141 330 3953; Fax: +44 141 942 6521; E-mail: o.sansom@beatson.gla.ac.uk
    †These authors contributed equally to this study

initiation. These include microtubule binding, mitosis and actin cytoskeleton regulation (Nathke, 2004).

β-catenin is also an essential part of the adherens junction at the membrane where it binds to E-cadherin. Loss of β-catenin from the intestine therefore leads not only to a loss of Wnt signalling (and stem cells) but also to reduced adhesion alongside cell loss from the villus (Ireland *et al*, 2004). The inter-relationship between β-catenin:E-cadherin and the β-catenin:TCF4 complexes has been a subject of much study and debate. Loss of E-Cadherin in cancer cells is often associated with markers of "EMT" and an upregulation of β-catenin TCF4 signalling, which is thought to happen at late stages of tumour progression.

However, despite this, there is a lack of definitive *in vivo* evidence that E-cadherin levels are able to limit the transforming capacity of β-catenin accumulation in tumour initiation. Presumably, this is due to the fact that increases in free β-catenin would be degraded by the destruction complex.

Given the centrality of the APC tumour suppressor to CRC, we decided to use a genetic approach *in vivo* and *in vitro* to address the differences in Wnt activation by either mutation of β-catenin or mutation of the destruction complex (APC or GSK3). We have previously shown that genetic deletion of both copies of the *Apc* tumour suppressor (*Apc^fl/fl^*) rapidly generates a crypt-progenitor cell-like phenotype (CPC) within the intestinal epithelium, with cells failing to differentiate, retaining proliferative capacity and failing to migrate up the crypt-villus axis (Sansom *et al*, 2004). This is associated with a relocalisation of β-catenin to the nucleus and the expression of functionally important Wnt target genes such as *cMyc* (Sansom *et al*, 2007).

We therefore investigated a number of alternative approaches to deregulation of Wnt signalling within the intestinal epithelium and assessed functional and Wnt signalling outputs. Taken together, our data show that mutation of the destruction complex, either by loss of APC or by loss of GSK3, leads to a rapid Wnt deregulation with accumulation of β-catenin in the small intestine and the colon. Surprisingly, a single activating mutation of β-catenin takes much more time to develop a phenotype in the small intestine and is unable to transform the colon. We show that this is due to high E-cadherin levels in the colon, which act as a sink for the mutated β-catenin. Reduction in E-cadherin or mutation of both copies of β-catenin swamps E-cadherin and then leads to rapid transformation of both the small intestine and colon.

# Results

## *Apc* loss, *Gsk3* loss and homozygous mutation of β-catenin are each sufficient to induce a crypt-progenitor phenotype

Our previous data showed that *Apc* loss was sufficient to induce β-catenin activation in the mouse intestine. The resulting Wnt signalling deregulation is characterised by a crypt-progenitor cell (CPC) phenotype with high nuclear β-catenin, increased proliferation and upregulation of Wnt target genes in both the small intestine and colon (Sansom *et al*, 2004; Hinoi *et al*, 2007; Fig 1A).

To confirm our finding that inactivation of the destruction complex was sufficient to initiate transformation of the small

intestine and colon, we deleted GSK3 and assessed if it would drive a similar phenotype to that of *Apc* loss. There are two isoforms of GSK3, GSK3alpha (GSK3A) and GSK3beta (GSK3B), and data from ES cells suggested that in the absence of one isoform, the other can compensate (Doble *et al*, 2007).

We generated *AhCre^ER^ Gsk3alpha^fl/fl^ Gsk3beta^fl/fl^* mice, which upon induction with β-naphthoflavone and tamoxifen recombine in the crypts of the small intestine and to a lesser extent in the colon (Kemp *et al*, 2004). Complete genetic ablation of *Gsk3* produced a phenotype that recapitulated *Apc* deficiency within the intestinal epithelium (Figs 1A and EV1A). Six days after *Gsk3* deletion, the intestines adopted the CPC phenotype with nuclear localisation of β-catenin and an induction of Wnt target genes. This similarity to the *Apc* loss phenotype suggested that the main function of GSK3 within the intestine is to control Wnt signalling. *Apc* heterozygous mice develop intestinal tumours on the loss of the remaining copy of *Apc*. Given there are 2 different *GSK3* isoforms, 4 mutations would be required to produce a situation where Wnt would be deregulated. We asked, if only 1 allele of *GSK3* needed to be lost, whether these mice would undergo intestinal tumourigenesis. To do this, we generated mice lacking 3 alleles of *GSK3* (*AhCre GSK3alpha^fl/fl^ GSK3beta^fl/+^* and *AhCre Gsk3alpha^fl/+^ Gsk3beta^fl/fl^* mice) and found following Cre deletion a significant fraction of these mice spontaneously developed intestinal tumours, in contrast to the single isoform mice (*AhCre Gsk3beta^fl/fl^*; Fig EV1B and C).

Similarly to loss of the destruction complex, activating mutation of both copies of the β-catenin allele (*AhCre^ER^ Catnb^lox(ex3)/lox(ex3)^*) showed the CPC phenotype with nuclear localisation of β-catenin and increased proliferation along the crypt-villus axis (Figs 1A and EV2A). Exon 3, which encodes the GSK3 phosphorylation sites that target β-catenin for degradation, is flanked by loxP sites, so the expression of Cre recombinase within the intestinal epithelium results in deletion of exon 3 and hence, β-catenin can no longer be targeted for degradation (Harada *et al*, 1999).

We investigated the impact of a single β-catenin mutation, which is analogous to the situation found in human patients. In contrast to *Apc* deletion (*AhCre^ER^ Apc^fl/fl^*), activation of a single copy of β-catenin did not yield a robust CPC phenotype at day 5 or day 10. Immunohistochemical staining of β-catenin showed very little nuclear localisation and no increase in proliferation (Figs 1B and EV2A).

At day 15, nuclear β-catenin became evident in *AhCre^ER^ Catnb^lox(ex3)/+^* mice and a number of crypts became enlarged. At time points past day 20, a single copy of *Catnb^lox(ex3)/+^* was able to induce a robust CPC phenotype in the small intestine (Fig 1B).

To probe the kinetics of β-catenin activation, we used intestinal "organoid" crypt culture (Sato *et al*, 2009). Previous studies have shown that once *Apc* is lost, intestinal cultures no longer require the addition of R-spondin or Wnt ligand produced by the Paneth cells. They also no longer bud into "crypt-like" structures, but form sphere structures instead (Fig 1C). The cultures therefore provide an excellent system to compare β-catenin exon 3 deletion to the loss of *Apc*. We therefore examined the capacity of *AhCre^ER^ Catnb^lox(ex3)/+^* and *AhCre^ER^ Catnb^lox(ex3)/lox(ex3)^* crypts (from the small intestine) which were induced *in vivo* to form organoids/ spheres in culture and their R-spondin1 dependence. In contrast to organoids where *Apc* was deleted, *AhCre^ER^ Catnb^lox(ex3)/+^* crypts did

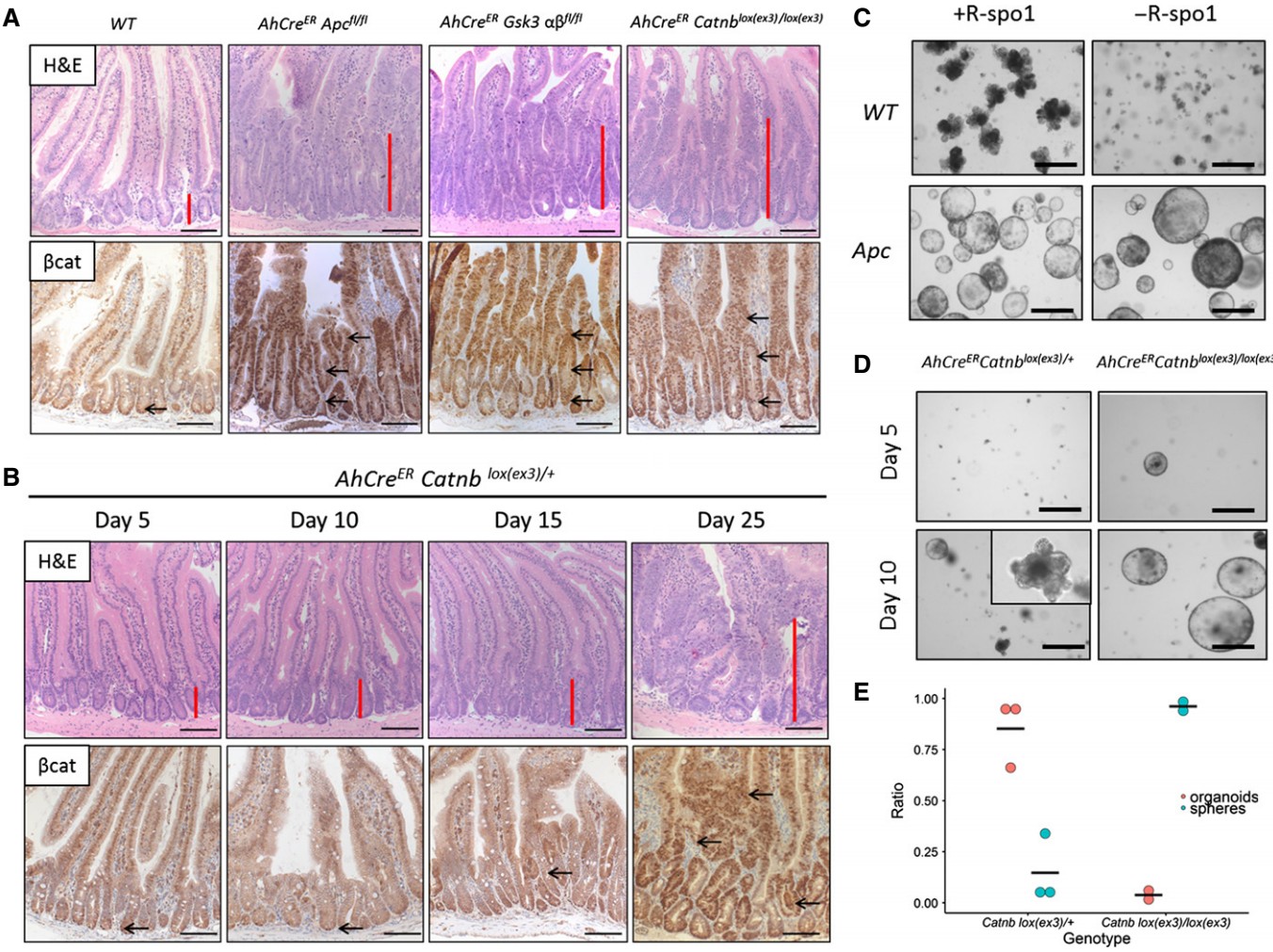

**Figure 1.** ***Apc* loss, GSK3 loss and homozygous mutation of β-catenin are sufficient to induce a rapid crypt-progenitor phenotype, but not a single β-catenin mutation.**

A Wild-type mice have small defined crypts in the small intestine with little nuclear β-catenin at the bottom of the crypt. The small intestine of *AhCre*[ER] *APC*[fl/fl], *AhCre*[ER] *Catnb*[lox(ex3)/lox(ex3)] (day 5) or *AhCre*[ER] *Gsk3alpha*[fl/fl] *beta*[fl/fl] (day 6) show the crypt-progenitor cell (CPC) phenotype with increased crypt size (red bar) and nuclear β-catenin (arrows) along the crypt-villus axis. Scale bar, 100 μm.

B A heterozygous activation of β-catenin (*AhCre*[ER] *Catnb*[lox(ex3)/+]) shows no increase in crypt size or nuclear β-catenin at days 5–10. At day 15 post-induction, accumulation of nuclear β-catenin (arrows) becomes evident with a dramatic CPC phenotype at about day 25. Scale bar, 100 μm.

C Culture of small intestinal crypts of WT and *VilCre*[ER] *Apc*[fl/fl] (or *AhCre*[ER] *Apc*[fl/fl]) mice with/without R-spo1 shows the dependence of the Wnt agonist in WT organoids but not in *Apc*-deficient spheres. Representative photos were taken at day 5 in culture. Black scale bar, 50 μm.

D At day 5 post-induction, only crypts from *AhCre*[ER] *Catnb*[lox(ex3)/lox(ex3)] but not *AhCre*[ER] *Catnb*[lox(ex3)/+] survive in culture without the addition of R-spo1. At day 10 post-induction, we observed a mixed phenotype of more organoid-like structures in *AhCre*[ER] *Catnb*[lox(ex3)/+] compared to spheres from *Catnb*[lox(ex3)/lox(ex3)] crypts in the first week of culture. Black scale bar, 50 μm.

E Quantification of organoids/spheres at day 10 post-induction (N = 2 or 3 mice per genotype, mean of 6 technical replicates per mouse).

not show R-spondin1 independence and died in culture when plated 5 days following Cre recombination *in vivo* (Fig 1D). In contrast, *AhCre*[ER] *Catnb*[lox(ex3)/lox(ex3)] formed spheres in an equivalent manner to homozygous *Apc* deletion, independent of R-spondin1. Crypts sampled at later time points (day 10) from *AhCre*[ER] *Catnb*[lox(ex3)/+] mice were able to survive in culture without R-spondin. At this time point, we observed a more organoid-like structure phenotype (~85%) than the expected typical round spheres (~15%) (Fig 1E). This early intermediate phenotype of crypts suggests that absolute levels of Wnt signalling in crypts with one copy of β-catenin were

not as high as either mutation of both alleles of β-catenin or *Apc* loss. In contrast, crypts from *AhCre*[ER] *Catnb*[lox(ex3)/lox(ex3)] almost completely formed spheres, and only few organoid-like structures were observed.

**Single copy activation of β-catenin does not transform the colon either immediately or at longer time points**

We next examined the colons from the *AhCre*[ER] *Catnb*[lox(ex3)/+] mice. In contrast to *AhCre*[ER] *Apc*[fl/fl] mice, which rapidly develop a CPC

phenotype with nuclear β-catenin accumulation and Wnt target gene activation, this did not occur in the $AhCre^{ER}$ $Catnb^{lox(ex3)/+}$ mice. Colons from $AhCre^{ER}$ $Catnb^{lox(ex3)/+}$ mice even 25 days post-induction did not show nuclear β-catenin or Wnt target gene activation despite a clear CPC phenotype in the small intestine (Fig EV2B).

To examine this phenomenon more carefully, we used the $VilCre^{ER}$ transgene to drive recombination as this delivers much higher recombination in the colorectal epithelium than the $AhCre^{ER}$ transgene. Four days post-recombination, a CPC phenotype was observed in the colons (and the small intestine) of $VilCre^{ER}$ $Catnb^{lox(ex3)/lox(ex3)}$ and $VilCre^{ER}$ $Apc^{fl/fl}$ mice with increased levels of nuclear β-catenin and SOX9. However, no phenotype or accumulation of β-catenin or Sox9 was observed in the colons (or small intestines) of $VilCre^{ER}$ $Catnb^{lox(ex3)/+}$ mice at this time point (Fig EV3). The dramatic CPC phenotype in the SI and colon meant both the $VilCre^{ER}$ $Catnb^{lox(ex3)/lox(ex3)}$ and $VilCre^{ER}$ $Apc^{fl/fl}$ needed to be harvested due to signs of ill health. In contrast, $VilCre^{ER}$ $Catnb^{lox(ex3)/+}$ developed signs of ill health after about 25 days which was associated with a small intestinal CPC phenotype. In the colon, there was not an obvious CPC phenotype at this stage (Figs EV2C and EV3).

Given this failure to drive a CPC phenotype, we next wanted to ask whether single copy activation of β-catenin within intestinal stem cells could lead to intestinal tumourigenesis. To do this, we interbred $Catnb^{lox(ex3)/+}$ to $Lgr5Cre^{ER}$ mice to allow inducible Cre-mediated recombination within the LGR5-positive intestinal stem cells. The $Lgr5Cre^{ER}$ delivers highly penetrant recombination in the small intestinal Lgr5-positive stem cell population and lower penetrant recombination in colon Lgr5-positive stem cell population. We have previously shown that targeting $Apc$ loss to the Lgr5 compartment led to rapid small intestinal adenoma formation (Barker *et al*, 2009; Myant *et al*, 2013). $Lgr5Cre^{ER}$ $Catnb^{lox(ex3)/+}$ and $Lgr5Cre^{ER}$ $Catnb^{lox(ex3)/lox(ex3)}$ mice were induced with a single intraperitoneal injection (IP) injection of tamoxifen and rapidly developed small intestinal tumours (Appendix Fig S1) with similar kinetics to those following $Apc$ deletion in the Lgr5 compartment. The equal potency of $Apc$ loss and mutation of a single β-catenin allele at transforming Lgr5-positive cells was somewhat surprising given the high frequency of $APC$ mutations in human CRC. Therefore, we next analysed colonic lesions within $Lgr5Cre^{ER}$ $Catnb^{lox(ex3)/+}$, $Lgr5Cre^{ER}$ $Catnb^{lox(ex3)/lox(ex3)}$ and $Lgr5Cre^{ER}$ $Apc^{fl/fl}$ mice when they were harvested due to small intestinal disease burden. We found that the majority of $Lgr5Cre^{ER}$ $Catnb^{lox(ex3)/lox(ex3)}$ mice (88%) and $Lgr5Cre^{ER}$ $Apc^{fl/fl}$ mice (91%) had lesions in the colon, whereas none of the $Lgr5Cre^{ER}$ $Catnb^{lox(ex3)/+}$ mice (0%) had any lesions (Fig 2A). Colonic lesions were defined by an increase in nuclear β-catenin levels and corresponding levels of the Wnt target gene Sox9 protein (Fig 2B). Given that we observe a similar accumulation of β-catenin in the small intestine after GSK3 deletion; we predicted that $Lgr5Cre^{ER}$ $Gsk3alpha^{fl/fl}$ $Gsk3beta^{fl/fl}$ should develop colonic lesions. We found that mice similarly succumbed to intestinal lesions in both the small intestine and colon (Fig EV4).

Taken together, this showed that there are fundamental differences in the ability of a single copy of β-catenin to transform small intestinal and colorectal epithelial cells in the mouse.

## E-cadherin levels are higher in the colon compared to the small intestine of wild-type mice

Although many studies have proposed that the pool of E-cadherin may limit the amount of free β-catenin, there is very little experimental evidence that this is the case. For example, E-cadherin heterozygous mice are viable and fertile with no clear deregulation of β-catenin activity (Larue *et al*, 1994; Riethmacher *et al*, 1995).

Given the clear phenotypical differences between the small and the large intestine in our $VilCre^{ER}$ $Catnb^{lox(ex3)/+}$ mice, we first examined whether E-cadherin levels might be higher in the colon than the small intestine. This would suggest that E-cadherin might be playing a role in limiting the free β-catenin. Investigating expression by both qRT–PCR and immunoblotting showed markedly higher levels of E-cadherin in the colon compared to the small intestine in wild-type mice (Fig 3A–C).

To quantify not just the E-cadherin levels in both tissues, but rather the complexes of β-catenin and E-cadherin, we used the well-characterised proximity ligation assay (PLA). Using antibodies for β-catenin and E-cadherin allowed us to quantify the number of complexes in the small intestine and the colon of wild-type mice. Consistent with the increase in E-cadherin expression, there was a 2- to 3-fold increase in the number of E-cadherin:β-catenin complexes in the colon compared to the small intestine (Fig 3D).

## Accumulation of mutated β-catenin at the adherens junctions is slower in the colon

Next, we investigated the dynamics of E-cadherin:β-catenin complexes. We were able to do this as deletion of exon 3 of β-catenin (and hence activation) produces a smaller protein which is easily detected by immunoblotting. Thus, in a $VilCre^{ER}$ $Catnb^{lox(ex3)/lox(ex3)}$ mouse, all newly produced β-catenin in intestinal epithelial cells will be smaller (Δex3). Using co-immuno-precipitation for E-cadherin, the relative amounts of mutant β-catenin complexed with E-cadherin can be assessed. We induced $VilCre^{ER}$ $Catnb^{lox(ex3)/lox(ex3)}$ mice and sampled the small intestine and the colon after 24 and 48 h. After just 24 h (Fig 4A, day 1), approximately 70% per cent of the β-catenin bound to E-cadherin is of the mutated form in the small intestine (72:28 Δex3:wt). In contrast, the majority of β-catenin in the colon bound to E-cadherin is of the WT form (35:65 Δex3:wt). By 48 h (day 2), in the small intestine (SI), approximately 90 per cent of β-catenin bound to E-cadherin is of the mutant form (90:10 Δex3:wt); similarly, the ratio in the colon shifted towards the mutant form (63:37 Δex3:wt). Thus, this shows that the E-cadherin:β-catenin complexes in the small intestine are rapidly saturated with mutant β-catenin.

We next analysed the ratio in $VilCre^{ER}$ $Catnb^{lox(ex3)/+}$ mice. Here, only half of the newly produced β-catenin will be of the mutant form, so we hypothesised that saturation would take longer, especially given the protracted time taken to generate the CPC phenotype. However, we would expect the mutant form to accumulate as this would not be turned over by the destruction complex. First, we examined E-cadherin:β-catenin complexes 4 days post-Cre induction. We showed that 75% of the β-catenin bound to E-cadherin is of the mutated form in the small intestine (75:25 Δex3:wt), whereas the colon has the mutant and the wild-type β-catenin in a roughly similar ratio (53:47 Δex3:wt). At day 8 after induction,

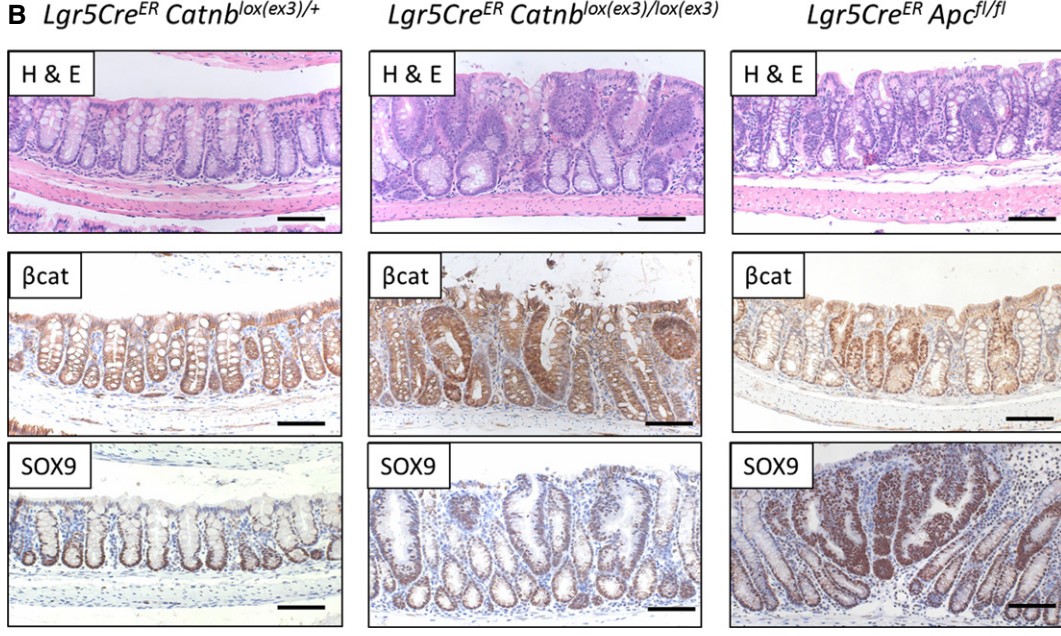

**A** Cohort of *Lgr5Cre^ER* mice aged until signs of intestinal tumour burden.

| Genotype | *Lgr5Cre^ER Catnb^lox(ex3)/+* | *Lgr5Cre^ER Catnb^lox(ex3)/lox(ex3)* | *Lgr5Cre^ER Apc^fl/fl* |
|---|---|---|---|
| No. of mice with colonic lesions | 0/15 | 8/9 | 11/12 |

**Figure 2.  Activation of one allele of β-catenin in Lgr5-positive stem cells does not result in colonic lesions.**

A    Table shows that only homozygous activation of β-catenin (*Lgr5Cre^ER Catnb^lox(ex3)/lox(ex3)*) or loss of *Apc* (*Lgr5Cre^ER Apc^fl/fl*) in Lgr5-positive stem cells results in colonic lesions in cohorts, sampled when signs of sickness were apparent. There were no lesions observed when only one allele of β-catenin (*Lgr5Cre^ER Catnb^lox(ex3)/+*) was mutated (chi-squared test, *P* < 0.001).

B    Colonic lesions were scored from H&E by identification of disorganised epithelium. Staining for β-catenin and SOX9 confirmed activation of Wnt signalling in the observed lesions. Scale bar, 100 μm.

approximately 85% of β-catenin bound to E-cadherin in the SI is mutated (85:15 Δex3:wt) compared to 55:45 Δex3:wt in the colon. Therefore, this relative increase in the mutant form of β-catenin compared to the wild-type protein showed that mutant β-catenin was specifically accumulating with E-cadherin at the adherens junction. This suggested that this could be acting as a sink for β-catenin, stopping the mutant β-catenin being translocated into the nucleus where it would bind TCF/LEF transcription factors and drive Wnt targets and the CPC phenotype.

**Haploinsufficiency for E-cadherin is sufficient for Wnt pathway activation in the presence of single allele β-catenin mutation**

If E-cadherin was acting as a sink and limiting mutant β-catenin from entering the nucleus, this would predict the following: (i) when β-catenin can be targeted for destruction, reduction in E-cadherin levels should have no phenotype, (ii) when there is mutant β-catenin (Δex), reduction in E-cadherin should cause this sink to be saturated quicker and hence lead to a translocation of β-catenin to the nucleus and a CPC phenotype.

To test this hypothesis, we intercrossed *AhCre^ER Catnb^lox(ex3)/+* mice to mice carrying a conditional knockout E-cadherin allele to generate *AhCre^ER Catnb^lox(ex3)/+ Cdh1^fl/+* mice and controls

(Boussadia *et al*, 2002). First, we examined the phenotype of loss of a single copy of E-cadherin. Five days following Cre induction, mice heterozygous for E-cadherin (*AhCre^ER Cdh1^fl/+* or *VilCre^ER Cdh1^fl/+*) showed a reduction in E-cadherin expression to ~60% and importantly a reduction in E-cadherin:β-catenin complexes assessed by PLA (Appendix Fig S2A and B). However, despite this reduction, intestines from these mice showed no phenotype and no increase in Wnt signalling (Fig EV5A). This is consistent with previous studies on whole body knockout E-cadherin heterozygotes where there are also no intestinal phenotypes (Larue *et al*, 1994; Riethmacher *et al*, 1995).

Finally, we investigated the phenotype of the *AhCre^ER Catnb^lox(ex3)/+ Cdh1^fl/+* mice. In contrast to the *AhCre^ER Catnb^lox(ex3)/+* 10 days post-induction, *AhCre^ER Catnb^lox(ex3)/+ Cdh1^fl/+* showed a CPC phenotype in both the small and large intestine with nuclear β-catenin and increased expression of the Wnt target Sox9 (Fig 5A and B). Indeed, a CPC phenotype with increased proliferation was already observed in the small intestine by day 5 (Fig EV5). The phenotype was confirmed using *VilCre^ER Catnb^lox(ex3)/+ Cdh1^fl/+* mice which had a CPC phenotype 4 days post-induction with increased expression of several Wnt target genes (*Axin2, Lgr5, cMyc, CD44*, Appendix Fig S3). Using the intestinal organoid system described earlier, small intestinal crypts from *AhCre^ER Catnb^lox(ex3)/+*

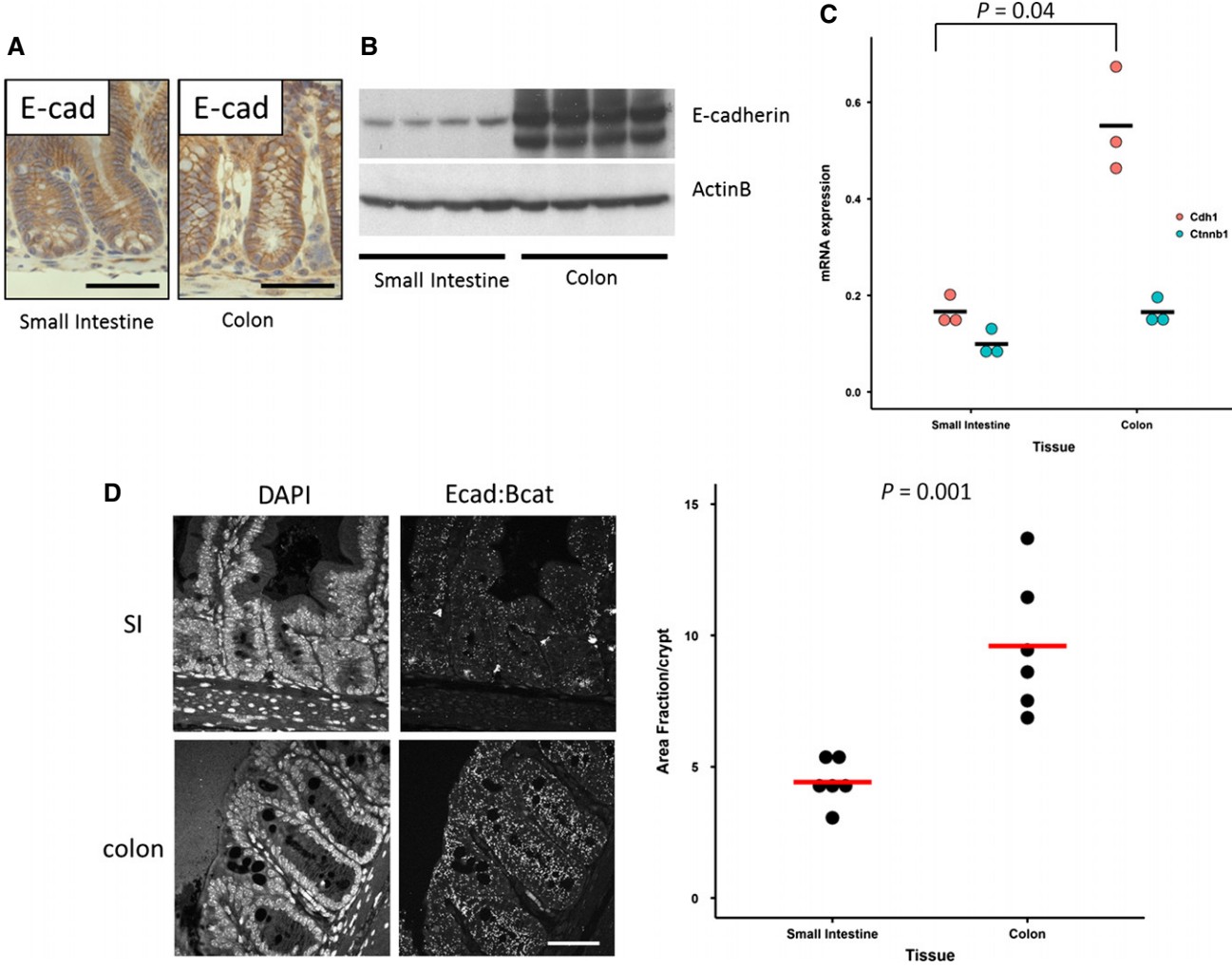

**Figure 3.  Increased E-cadherin:β-catenin levels in colonic crypts of wild-type mice.**

A   Staining of small intestinal and colonic crypts for E-cadherin. Scale bar, 100 μm.

B   Western blot of purified crypts from the small intestine and colon (N = 4 mice).

C   qRT–PCR of whole tissue from small intestine and colon. Expression of mRNA ($2^{(-dCt)}$) calculated relative to GAPDH (N = 3 mice). Statistics: one-sided Mann–Whitney U-test.

D   Proximity ligation assay for E-cadherin and β-catenin. For each mouse (N = 6), at least 10 crypts per tissue were analysed to calculate the mean. Statistics: one-sided Mann–Whitney U-test; white scale bar, 50 μm.

$Cdh1^{fl/+}$ mice formed spheroids in an R-spondin1-independent manner 5 days post-induction (Fig 5C). This was in contrast to $AhCre^{ER}$ $Catnb^{lox(ex3)/+}$ mice. Moreover, targeting a copy of E-cadherin loss in combination with a β-catenin mutation to Lgr5 ISC ($Lgr5Cre^{ER}$ $Catnb^{lox(ex3)/+}$ $Cdh1^{fl/+}$) significantly accelerated tumourigenesis when compared to $Lgr5Cre^{ER}$ $Catnb^{lox(ex3)/+}$ mice. Most importantly, when these mice with reduced E-cadherin were analysed for colonic lesions, 6 of 7 mice had lesions with high nuclear β-catenin (Fig 5D).

It is interesting to note that when attempting to recapitulate our crypt culture findings from $AhCre^{ER}$ $Catnb^{lox(ex3)/+}$ mice (Fig 5C) using colonic crypts, we found that the colonic crypts grew more successfully than the ones from the small intestine despite the fact that *in vivo* β-catenin mutation could not transform the colon (Appendix Fig S5A). A possible explanation for this is the different EDTA concentration needed to purify the crypts from the small

intestine and the colon (2 and 25 mM EDTA, respectively) which is known to disrupt the E-cadherin bindings. Indeed, when we analysed wild-type crypts straight after the EDTA purification, we observe similar PLA counts for the crypts of both tissues (Appendix Fig S5B). These changes persist *in vitro*, when we compared intestinal crypts from $VilCre^{ER}$ $Apc^{fl/fl}$ from the small intestine and the colon (Appendix Fig S6A and B).

**Human cancers, characterised by β-catenin mutation, are associated with reduction in E-cadherin levels**

We finally wanted to examine if these observations from the mouse might possibly translate to human tumourigenesis. Recent sequencing studies with over 200 CRC showed that mutations in β-catenin (*CTNNB1*) in colorectal cancer are very rare, approximately 5 % (11/212). Closer analysis showed that many of

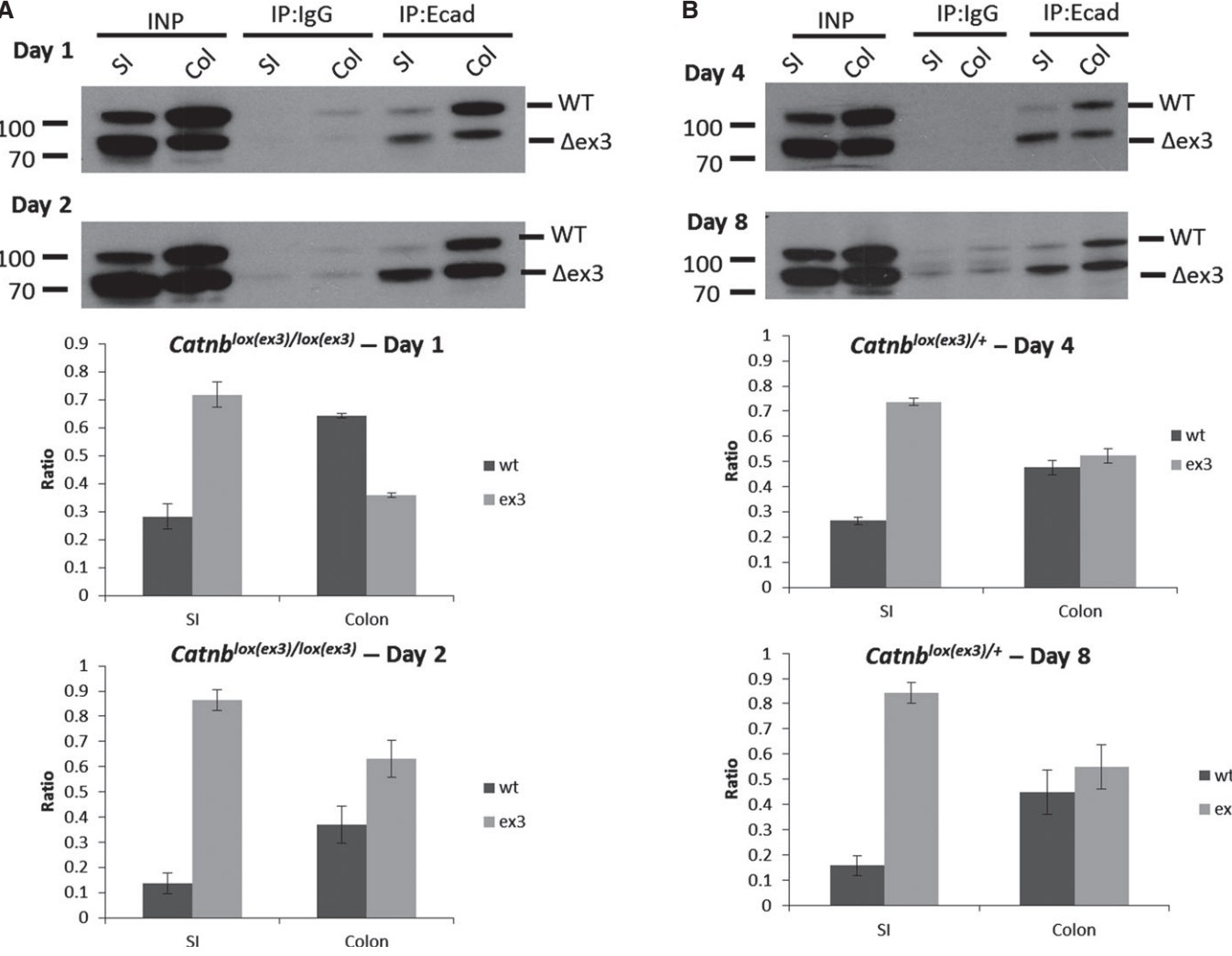

**Figure 4.  E-cadherin saturates with mutant β-catenin over time.**

A   Immunoprecipitation (IP) of E-cadherin from *VilCre^ER Catnb^lox(ex3)/lox(ex3)* mice, 24 h (day 1, top) and 48 h (day 2, bottom) after induction. The ratio of wild-type (WT, top lane) and mutant exon 3 β-catenin (Δex3, bottom lane) bound to E-cadherin (IP:Ecad) was calculated for each tissue. Graphs show average of experiments (*N* = 3 mice for each genotype and time point); error bars represent s.e.m.

B   IP of E-cadherin from *VilCre^ER Catnb^lox(ex3)/+* mice at day 4 and day 8. Graphs show average of experiments (*N* = 3 mice for each genotype and time point); error bars represent s.e.m.

these were not in exon 3 and therefore unlikely to affect β-catenin degradation. In total, only 2 of 11 mutations were in the most common exon 3 hotspot (codons 31–45; Appendix Fig S4). Comparisons of these mutations with cancers where activating mutations of *CTNNB1* are common such as hepatocellular carcinoma (HCC) or solid pseudopapillary tumours of the pancreas show very different patterns of mutations (cbioportal, Cerami *et al*, 2012; Gao *et al*, 2013), indicating these maybe passenger mutations (Appendix Fig S4D).

Given almost all solid pseudopapillary tumours (SPT) have a β-catenin mutation in exon 3, this suggests deregulation of Wnt signalling as an essential step for tumourigenesis in this cancer. From our model, we would predict a change in E-cadherin levels for a β-catenin activating mutation to have the greatest transforming properties. Importantly in these SPT tumours, in addition to the nuclear accumulation of β-catenin, an aberrant localisation of E-cadherin has

been reported (Chetty & Serra, 2008). We therefore analysed a tissue microarray (TMA) with normal and SPT tissue for E-cadherin: β-catenin complexes and saw a dramatic decrease in the number of complexes in SPT tumours, compared to normal tissue (Fig 6A).

Finally, we examined hepatocellular carcinomas (HCC) where approximately 20% of tumours have a mutation in exon 3 of β-catenin. Analysing expression data from 269 patients (TCGA provisional), we saw a significant negative correlation of E-cadherin with several Wnt target genes (Fig 6B).

## Discussion

Although APC is well established in controlling the Wnt signalling pathway, there has been controversy over the role of the activation of Wnt signalling in the initiation of CRC. Much of this debate has

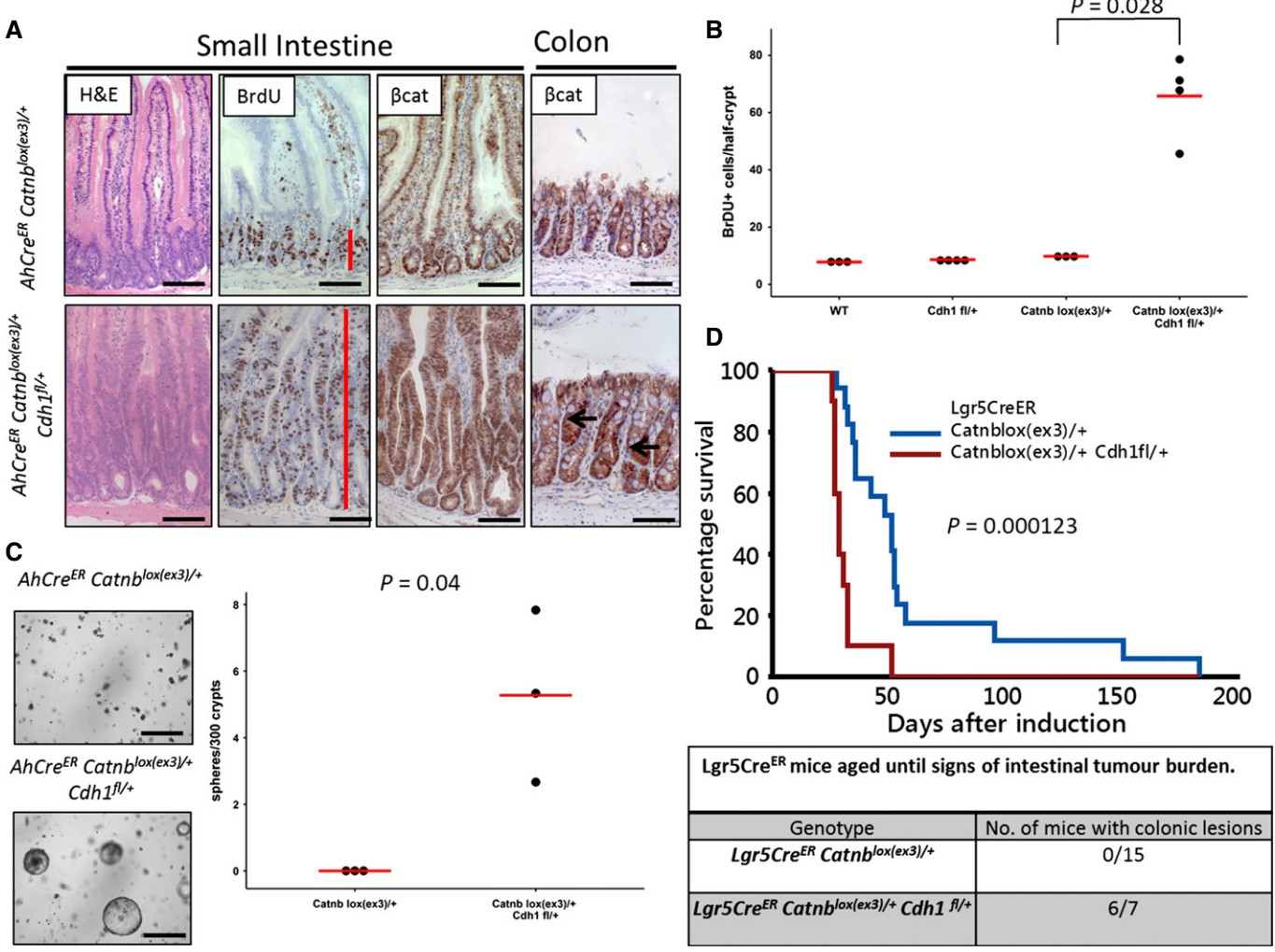

**Figure 5.    Haploinsufficiency for E-cadherin in the presence of single allele β-catenin mutation leads to Wnt deregulation.**

A    *AhCre^ER Catnb^{lox(ex3)/+}* compared to *AhCre^ER Catnb^{lox(ex3)/+} Cdh1^{fl/+}* at day 10 post-induction. Note the presence of colonic lesions in the colon of *AhCre^ER Catnb^{lox(ex3)/+} Cdh1^{fl/+}* mice (arrows). Scale bar, 100 μm; red bar indicates proliferative zone (BrdU).

B    Scoring of BrdU-positive cells per half-crypt in the small intestine of wild-type, *AhCre^ER Cdh1^{fl/+}*, *AhCre^ER Catnb^{lox(ex3)/+}* and *AhCre^ER Catnb^{lox(ex3)/+} Cdh1^{fl/+}* mice at day 10 post-induction. $N \geq 3$, at least 25 crypts per mouse were scored. There was significantly higher proliferation in the *AhCre^ER Catnb^{lox(ex3)/+} Cdh1^{fl/+}* mice ($P = 0.028$, one-sided Mann–Whitney U-test).

C    *In vitro* growth of crypts (small intestine) from mice as indicated at day 5 post-induction without R-spo1. Quantification of sphere-forming efficiency of *AhCre^ER Catnb^{lox(ex3)/+} AhCre^ER Catnb^{lox(ex3)/+} Cdh1^{fl/+}*, day 5 post-induction. $N = 3$ mice per genotype, mean of 6 technical replicates per mouse, $P = 0.04$ one-sided Mann–Whitney U-test.

D    Survival of *Lgr5Cre^ER Catnb^{lox(ex3)/+}* and *Lgr5Cre^ER Catnb^{lox(ex3)/+} Cdh1^{fl/+}* shows significant acceleration ($P = 0.000123$, log-rank test) after E-cadherin reduction. About 85% (6/7 mice) had colonic lesions, as identified with β-catenin IHC in contrast to *Lgr5Cre^ER Catnb^{lox(ex3)/+}* mice.

been due to the failure to detect nuclear β-catenin at the early stages of the disease and its relatively heterogeneous nature at later stages. Moreover, data from experiments in zebrafish have suggested other pathways to be very important for the phenotypes of Apc loss (Phelps *et al*, 2009). This coupled with the rarity of β-catenin mutations (after early studies suggesting much higher rates (Sparks *et al*, 1998)) and the lack of mutual exclusivity with *APC* mutations has led to many discussions on the reasons for *APC* mutation in CRC.

Here, we provide definitive *in vivo* proof (in the murine intestine) that activation of Wnt signalling by loss of the destruction complex or bi-allelic β-catenin mutation is sufficient to drive rapid intestinal

transformation. In contrast, upon monoallelic deletion of exon 3, the mutant form of β-catenin takes much longer to accumulate in the small intestine and lead to a CPC phenotype. In the colon, monoallelic mutation of β-catenin did not drive a CPC phenotype (even 25 days post-induction) and did not cause lesions when targeted to the Lgr5 compartment. This suggests that the level of Wnt deregulation required to transform the colorectal epithelium is higher than in the small intestine. This idea often referred to the "just right" model has previously been associated with different levels of Wnt signalling induced by different *APC* mutations (Fodde & Brabletz, 2007; Buchert *et al*, 2010). Moreover, within the *Apc^Min* mouse, the distribution of tumours in the small and large

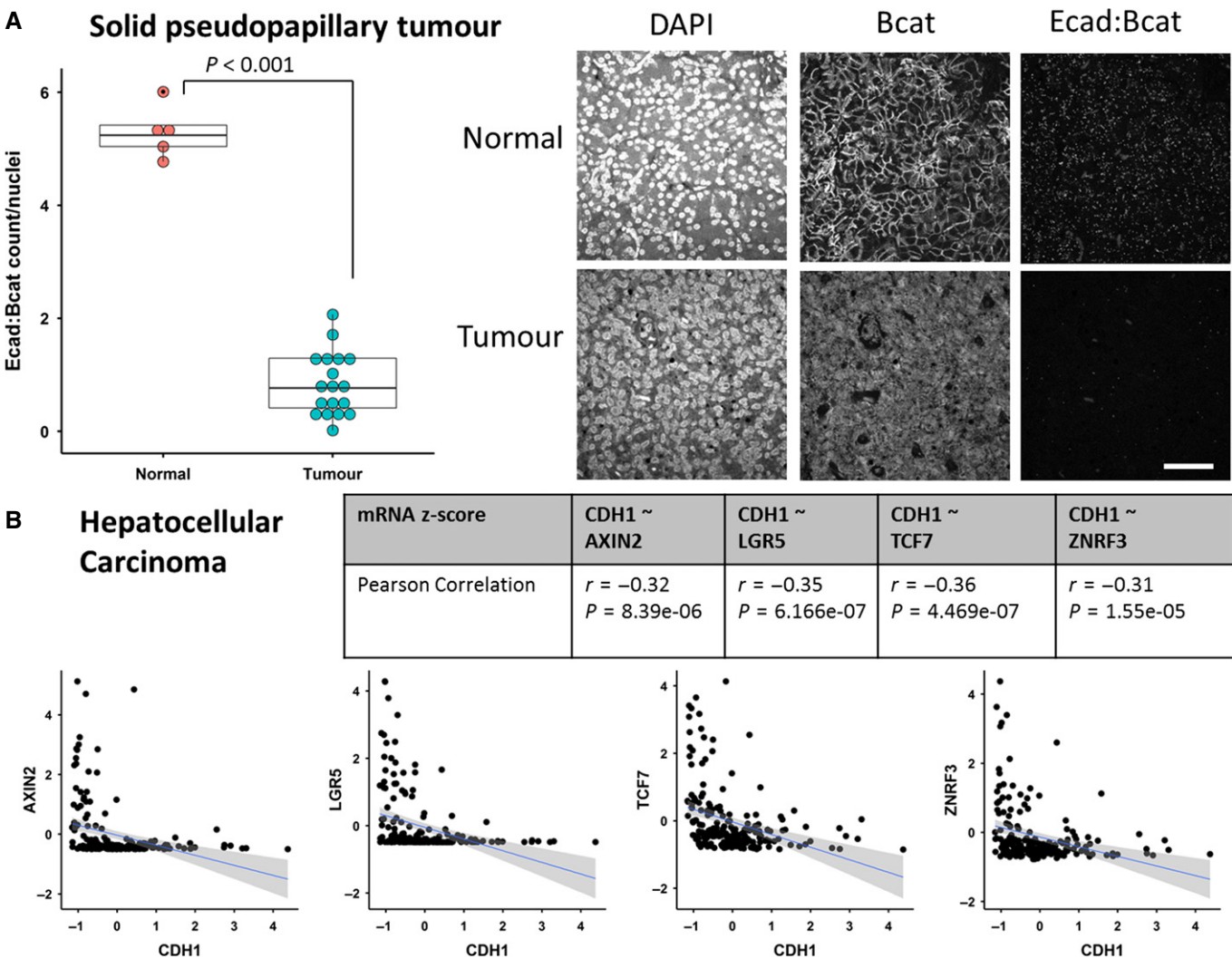

**Figure 6.** **Human cancers, characterised by β-catenin mutation, are associated with reduction in E-cadherin levels.**

A Proximity ligation assay for E-cadherin:β-catenin on a tissue microarray of SPT patients. PLA dots/cells in normal and tumour tissue were counted. Each dot in the boxplot represents the average for a single patient. Staining for β-catenin confirms accumulation of nuclear β-catenin in tumour tissue. Representative pictures are shown. N = 18, statistics: Mann–Whitney U-test, P = 0.0009051. White scale bar, 50 μm.

B Correlation of several Wnt target genes with the expression of E-cadherin (CDH1) was analysed in 269 HCC patients (TCGA provisional), and linear regression line (blue) and confidence region (shaded) were added.

intestines has been suggested to reflect Wnt gradients within the tissue (Leedham *et al*, 2013). Differences between the colon and small intestine in Wnt signalling have been reported in mice deficient for Tcf-4. In this case, deletion of Tcf-4 stopped the formation of small intestinal crypts, but colonic crypts were still present (Korinek *et al*, 1998).

The differences between the small intestine and the colon correlated with higher E-cadherin levels and increased E-cadherin:β-catenin complexes in colonic crypts. We show that E-cadherin can act as a sink for mutated β-catenin and that the adherens junctions saturate over time with the mutant form of β-catenin. Importantly, a reduction in the expression of E-cadherin leads to a quicker accumulation of sufficient β-catenin to facilitate transformation (Fig 7). To our knowledge, this is the first demonstration *in vivo* that levels of E-cadherin can be limiting to cancer initiation. It is interesting to

note that mice heterozygous for E-cadherin (*Cdh1*$^{+/−}$) have an increased tumour burden when crossed to the *Apc*$^{1638N/+}$ mouse model, a long latency model of *Apc*-loss induced tumourigenesis (Smits *et al*, 2000).

When we examined solid pseudopapillary tumours of the pancreas, a tumour characterised by β-catenin mutations within exon 3, these tumours showed a strong reduction in E-cadherin:β-catenin complexes. Moreover in HCC, a tumour type which has approximately 20% β-catenin exon 3 mutations, there was a good correlation between reduction in E-cadherin and activation of Wnt signalling targets. Thus, it may be that in these cancers, E-cadherin limits the precise levels of Wnt signalling driven by β-catenin mutation. Thus, downregulation of E-cadherin in these tumours may drive tumour progression. It should also be noted that opposing patterns of Wnt signalling and E-cadherin have been shown in

murine liver, with β-catenin higher in zone 3 of the liver and E-cadherin in zone 1. Therefore, one might predict that β-catenin mutations would yield a greater phenotype in hepatocytes from zone 3 of the liver versus zone 1 (Benhamouche *et al*, 2006). Hence, one could speculate E-cadherin levels might modulate the cell of origin of HCC that carry a β-catenin mutation.

One key question from our work is how does this translate to human cancer? Why do humans not get small intestinal cancer with β-catenin exon 3 mutations rather than CRC with *APC* mutations? One obvious difference is in sporadic human CRC, tumours develop over a number of years and the sequential mutations in *APC* may provide different selective benefits and other functions of APC may be very important in human carcinogenesis. Data exist showing that stepwise *Apc* loss might also induce a more dramatic phenotype (Fischer *et al*, 2012). Moreover, it is also possible that some of the other consequences of APC mutation may only be revealed when there are only a small number of APC-deficient cells rather than entire crypts. For example, the Wnt-independent roles of APC in microtubules, centrosomes and mitosis have been suggested to modify stem cell division and could favour retention of APC-deficient ISC. We have observed clear differences between *Gsk3* and *Apc* deletion in the ability to respond to microtubule-stabilising drugs which show these other functions of APC are important *in vivo* (Radulescu *et al*, 2010).

One could also imagine different dietary carcinogens and microbiota leading to differential mutational spectra that might favour the observed loss of functions mutations in *APC* rather than the very

specific activating mutations required to produce non-degradable β-catenin.

Another potential reason for the lack of β-catenin mutations in both small intestinal and colon cancer is suggested from recent work investigating the selective advantage of different tumour-promoting mutations in murine small and large intestinal epithelium. The Winton group showed that a single inactivating *Apc* mutation in the intestine had an advantage over wild-type stem cells (Vermeulen *et al*, 2013). This increased the likelihood that a stem cell carrying this mutation would be retained to repopulate the entire crypt. A second mutation in *Apc* further increased the fitness of the stem cell compared to wild-type and was even more likely to repopulate the entire crypt. However, it is of interest to note that despite these selective advantages, given the increase in fitness never increased the probability of repopulating crypt to 1 (WT is neutral 0.5, the $Apc^{+/-}$ 0.62 and $Apc^{-/-}$ 0.79), then on many occasions, cells carrying *Apc* mutations would be lost. From our work, we would suggest that as it takes a protracted time for the β-catenin exon 3 mutation to have a phenotype in the small intestine, this mutation would probably be neutral (0.5) for some time. Thus, given neutral drift within the intestine, one might predict that these mutant cells would be lost.

For many years, there has been much discussion of why $Apc^{Min}$ mice (a model of FAP) mainly develop small intestinal tumours rather than FAP where patients preferentially develop colon tumours, though they will eventually develop duodenal tumours. Our data show acute Apc loss manifests phenotypes in both the

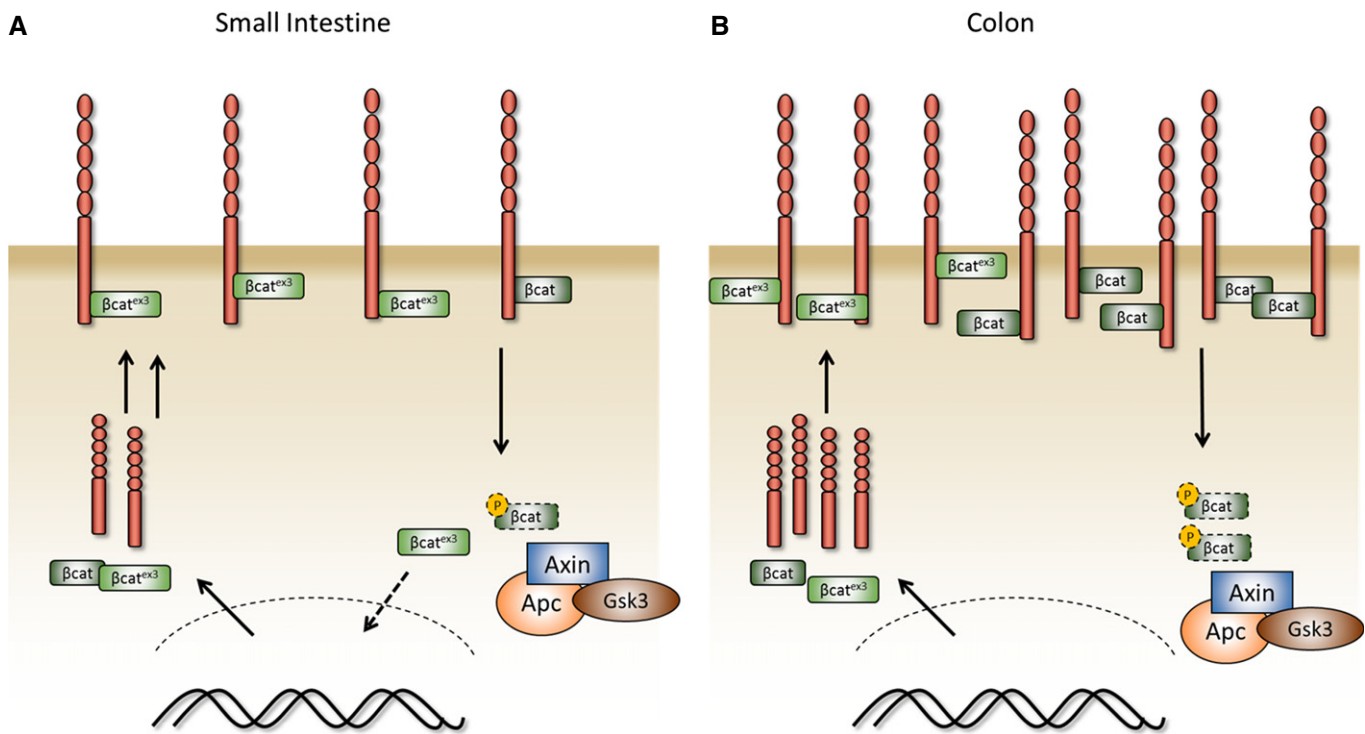

**Figure 7.  Model of single β-catenin mutation and interaction with E-cadherin in the small intestine and colon.**

A, B   Model of a single activating β-catenin mutation in the murine small intestine and the colon. In contrast to the small intestine (A), the increased levels of E-cadherin in the colon complexed with mutant β-catenin prevent its accumulation in the nucleus (B).

small intestine and colon which does not explain the tropism for small intestine in the mouse. A recent study by Tomasetti and Vogelstein (Tomasetti & Vogelstein, 2015) has suggested that differences in stem cell proliferation within colon versus small intestine could explain the tropism to colon in the human CRC.

We have also shown that within the intestine, GSK3 is only obviously limiting to the Wnt signalling pathway. Given GSK3 has many functions within the cells and is highly expressed, this suggests other kinases can compensate grossly for these activities within the intestine. It has previously been postulated that GSK3 inhibition may also be of therapeutic relevance for diabetes. One obvious concern is that long-term GSK3 inhibition might predispose to colorectal cancer. Our data here and indeed from hypomorphic mutants in *Apc* (Buchert *et al*, 2010) argue that to induce the crypt-progenitor phenotype and tumourigenesis, complete gene function has to be deleted rather than being simply inhibited. Thus, as GSK3 inhibitors would only reduce rather than ablate GSK3, it is unlikely this would predispose to cancer. Moreover, given the fact that there are 4 alleles of *GSK3,* mutation of a single allele would not activate the Wnt pathway much further, so there would be no selection for loss of GSK3, reinforcing the notion that GSK3 inhibition would not drive CRC. Therefore, there may be a therapeutic opportunity for GSK3 in diabetes.

In summary, we show that a single β-catenin mutation is not as potent in activating the Wnt signalling pathway as loss of *Apc*. Furthermore, we show that E-cadherin levels can limit the transforming potential of an activating β-catenin mutation.

## Materials and Methods

### Mouse experiments

All experiments were performed under the UK Home Office guidelines. The alleles used in this study were as follows: *AhCre^{ER}* (Kemp *et al*, 2004) *VilCre^{ER}* (el Marjou *et al*, 2004), *AhCre* (Ireland *et al*, 2004), *Lgr5Cre^{ER}* (Barker *et al*, 2007) *Catnb1^{lox(ex3)}* (Harada *et al*, 1999), *Gsk3alpha^{fl}, Gsk3beta^{fl}* (Patel *et al*, 2008), *Apc^{fl}* (Shibata *et al*, 1997) and *Cdh1^{fl}* (Boussadia *et al*, 2002). Cre induction strategies are described in the Appendix Supplementary Methods. *N* numbers for the animal experiments are provided on the figure legends.

### Immunohistochemistry

Standard immunohistochemistry techniques were used throughout this study. The following primary antibodies were used: BrdU (1/200, #347580, BD Biosciences), pGSKB-Ser9 (1/200, #9336, Cell Signaling), β-catenin (1/50 #610154, BD Biosciences), cMyc (1/200, sc-764, Santa Cruz), Sox9 (1/500, #AB5535 Chemicon), lysozyme (1/200, DAKO #A0099) and E-cadherin (1/200, R&D Systems AF748). Staining for nuclear β-catenin was performed on tissue samples fixed at 4°C for less than 24 h in 10% formalin prior to processing.

### Immunoblotting

Standard Western blot techniques were used. Crypts from the small and large intestine were purified by incubation with 2 and 25 mM

EDTA/PBS (respectively) for 30 min at 4°C. Crypts were further purified by centrifugation with low speed (50 × *g*, 3 min). The following antibodies were used: E-cadherin (1/5,000, BD Transduction Lab, #610182); β-actin (1/5,000, Sigma A2228).

### Immunoprecipitation

Small intestinal and colonic tissue (~4 cm) was lysed in lysis buffer (200 mM NaCl, 75 mM Tris–HCl [pH 7], 7.5 mM EDTA, 7.5 mM EGTA, 0.15% [v/v] Tween-20). Lysates were clarified by centrifugation at 16,000 × *g* for 10 min at 4°C. Magnetic beads conjugated to anti-mouse IgG (Dynabeads) were incubated with 1 mg of protein lysates, 1 μg of either monoclonal anti-E-cadherin (BD Biosciences #610182) or IgG isotype control (Sigma) for 1 h at 4°C with constant rotation. After several washes with lysis buffer, bound proteins were eluted from the beads by boiling at 100°C for 5 min in SDS reducing buffer. Bound proteins and 5 μg of total lysates (Input) were run on 4–12% Bis–Tris Gradient SDS gel and probed for β-catenin (1/1,000 BD Biosciences, #610154).

### Proximity Ligation Assay (PLA)

PLA was performed on tissue samples fixed at 4°C for < 24 h in 10% formalin prior to processing using the Duolink Detection kit (Sigma) according to the manufacturer's instructions. Briefly, after citrate buffer-mediated antigen retrieval, the slides were incubated with goat E-cadherin (1/200, R&D Systems AF748) and mouse β-catenin (1/2,000 for mouse tissue, 1/200 for human tissue, #610154, BD Biosciences) overnight. Detection was performed with PLA probes (anti-goat and anti-mouse) conjugated to oligonucleotides. After ligation, amplification detection with a fluorescent probe, slides were imaged on a Zeiss LSM confocal microscope. *Z*-stacks with 40× objectives were taken. PLA dots in crypts were analysed with ImageJ and either calculated as area fraction or count/nuclei.

### Sphere culture

Crypts were purified from mice as previously described (Sato *et al*, 2009). Crypts were seeded in growth factor-reduced Matrigel (BD Biosciences) with the addition of EGF and Noggin (both Peprotech), without R-spo1 or with 50 ng/ml R-spo1 (R&D Systems). About 300 crypts were plated per 20 μl matrigel/well, and procedures were carried out within 3 h after sacrificing the mouse. Representative photographs were taken after 7 days.

### Human TMA of solid papillary tumours (SPT) of the pancreas

Tissue microarray of 18 SPT patients was used as described before, and normal tissue was used as control (Serra *et al*, 2007). PLA was performed as described above.

### qRT–PCR of mouse intestine

For detailed description and primer sequences, see Appendix Supplementary Methods.

**Expanded View** for this article is available online:
http://emboj.embopress.org

## Acknowledgements

The authors would like to thank James Woodgett for the supply of Gsk3 floxed mice and Trevor Graham for helpful comments. Grant support: supported by Cancer Research UK grant C596/A17196, and DJH is funded by the European Union Seventh Framework Programme FP7/2007–2013 under grant agreement number 278568. FS is funded by the North West Cancer Research Fund CR700. OJS is funded by an ERC investigator grant COLONCAN.

## Author contributions

DJH, RAR, ARC and OJS contributed to study concept and design; DJH RAR, SR, ML, ADC, SB, SL, GM, LP, JM, FS, FC, KRR, VSM, BD and NB to acquisition of data; DJH, RAR, SR, AH, FC, BD, AW, LL and OS to analysis and interpretation of the data; DJH, RAR and OJS to drafting of the manuscript; AH and GK to statistical analysis; and AM, SS and RC to provision of reagents.

## Conflict of interest

The authors declare that they have no conflict of interest.

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
