## [Review Process File · The EMBO Journal]

Manuscript EMBO-2015-91739

E-cadherin can limit the transforming properties of activating β -catenin mutations

David J. Huels, Rachel A. Ridgway, Sorina Radulescu, Marc Leushacke, Andrew D. Campbell, Sujata Biswas, Simon Leedham, Stefano Serra, Runjan Chetty, Guenievre Moreaux, Lee Parry, James Matthews, Fei Song, Ann Hedley, Gabriela Kalna, Fatih Ceteci, Karen R. Reed, Valerie S. Meniel, Aoife Maguire, Brendan Doyle, Ola Söderberg, Nicholas Barker, Alastair Watson, Lionel Larue, Alan R. Clarke

Corresponding author: Owen J. Sansom, Cancer Research UK Beatson Laboratories

Review timeline:

Submission date:	08 September 2014
Editorial Decision:	14 October 2014
Resubmission:	08 April 2015
Editorial Decision:	27 May 2015
Revision received:	23 June 2015
Accepted:	01 July 2015

Transaction Report:

Editors: Thomas Schwarz-Romond and David del Álamo

1st Editorial Decision

14 October 2014

Thank you very much indeed for considering The EMBO Journal for presentation of your work that aims to elucidate the basis of distinct mutational load in colorectal carcinogenesis, taking advantage of your broad expertise in mouse genetic engineering.

The paper has been evaluated both on the conceptual level as well as the amount of definitive molecular insight from three scientific experts. As you will see, ref#2 appears rather supportive, while both ref#1 and #3 bring up two rather critical points: (i) both infer that the concept of the 'just right' Wnt-signaling threshold for the onset of CRC-formation is a long -debated one. (ii) They also appreciate that your data suggest single beta-catenin mutations as not being sufficient to drive intestinal tumorigenesis and distinct E-cadherin level determining the overall signaling output (which appears yet again not entirely novel). They therefore demand novel and definitive advances into these questions, at least from the more general expectations at The EMBO Journal. (As an aside, I do find the suggested, comparative mass-spec study to search for distinct beta-catenin interactors in various tumor tissues rather intriguing). Similarly, ref#3 finds the E-cadherin threshold model appealing (yet again this is not in itself novel), but sees much further reaching and definitive investigation warranted. (With the organoid model, you may have the right tools at hand).

On face value, I hope you have to agree that these requests amount to significant experimental revisions with a completely uncertain outcome.

As you know, we run a policy to only invite revision for papers with a predictable and timely turn-around time. We therefore find it more prudent at this stage to return the paper to you to decide whether to take the study significantly further OR seek rapid publication with an adjusted version, a decision that should be based on thorough estimates of resources and general commitment to this particular project.

Please also note that our sister title EMBOreports is increasingly interested and open in for rapid manuscript transfers. As the editors there operate fully independent, I have to ask you to take this option up/consider transfer of the relevant referee comments for their information entirely on your own.

Please trust that I did not take this decision lightly, recognizing the amount and quality of submitted work. I am truly sorry to be unable to transmit more encouraging news.

REFeree REPORTS

Referee #1:

Review: Precise Wnt signaling output explain why Apc and not b-catenin mutations are common in Colon Cancer.

This manuscript aims to dissect the differential role of APC and b-catenin mutations in CRC (colorectal cancer). The authors present evidence that b-catenin mutations are rare in CRC and bi-allelic b-catenin activating mutations are required to drive Wnt transformation in the colon. They further suggest that a single activating mutation of b-catenin is not able to drive transformation due to high colonic levels of the b-catenin binding protein E-cadherin. Finally, the authors suggest that co-deletion of a single copy of E-cadherin in the colon synergizes with a mono-allelic b-catenin mutation to transform intestinal progenitor cells toward colon cancer. Unfortunately, the experimental design contains a number of flaws, and the conclusions drawn are not supported by the results as indicated as follows:

1. Data presented in Fig. 1 includes TCGA analysis, which the authors claim indicates that b-catenin mutations in CRC do not occur in exon 3. However, this exon contains the sites at which mutations have been shown to block phosphorylation and ubiquitination in CRC.
2. The authors fail to compare b-catenin levels target gene activation in APC vs b-catenin mutant tumors, which could help in testing their model.
3. A major weakness of the study is the complete lack of evidence that a single activating mutation of b-catenin is unable to drive Wnt transcriptional activity due to higher E-cadherin levels in the colon as compared to the small intestine. The authors do not provide any data supporting the proposed mechanism by which E-cadherin would sequester mutated b-catenin in the cytoplasm. The authors need to perform experiments to support this hypothesis as well as exclude different mechanisms involved in the lack of Wnt transcriptional activation by a single mutated b-catenin in the colon. This evidence is particularly critical, since b-catenin mutations in HCC or in pseudopapillary pancreatic carcinomas, which are mostly heterozygous, are sufficient to drive Wnt transcriptional activation and to participate in transformation as cited by the authors. The authors could employ mass-spec studies to identify differential b-catenin partners in CRC, HCC, and pseudopapillary pancreatic carcinomas. The authors could also apply the same comparison using the engineered CRC mouse models carrying +/+, lox(ex3)/+, and lox(ex3)/lox(ex3) b-catenin.
4. In figure 5, E-cadherin levels should be measured in other tissues where b-catenin is mutated such as in HCC and pseudopapillary pancreatic carcinomas to further support its role in regulating b-catenin localization.
5. IP studies would be needed to establish whether higher E-cadherin levels are able to sequester mutated b-catenin in the cytoplasm.

6. To assess Wnt transcriptional activation, the authors only tested (by IHC) a single target gene, which varies within different experiments presented (i.e. SOX9 in Figure 4 and c-MYC in Figure 7). The authors need to evaluate more than one Wnt target in each set of experiments and consistently test these same genes in all experiments.

7. The title is not supported by the results provided.

8. The authors need to test their conclusions from mouse studies by evaluating the Wnt transcriptional gene signature on laser captured human CRC samples harboring either APC or CTNNB1 mutations by qPCR or RNAseq analysis.

Referee #2:

The group has a longstanding experience in establishing and analysing Wnt dependent mouse models of intestinal tumorigenesis. Thus, from a technical point of view everything is done well and there is not much to criticise. Also, the results as such are clear-cut and well documented. However, it is the significance and interpretation of results which is a matter of criticism.

1 In particular conclusions drawn in the manuscript title that these studies explain why b-catenin mutations are rare in human colorectal cancer are too simple. Results in the small intestine show that APC deleted mice or homozygously mutated b-catenin mice develop lesions termed CPC already after 5 days whereas heterozygous b-catenin mice do so after 25 days. While these results suggest that it is the dose of b-catenin that determines whether transformation becomes visible they do also show that a single b-catenin mutation in principle suffices, it just takes a bit longer. Revealingly, when expressed in epithelial stem cells using Lgr5Cre there was no difference in CPC development between the three genotypes showing that a single b-catenin is sufficient in the specific cellular background. By extension the finding that the heterozygous b-catenin mice did not show phenotypes after 25 days in the colon could be due to the limited period of observation. The finding that mutation of E-cadherin leads to CPC in b-catenin heterozygous b-catenin mice just underscores the relevance of b-catenin dosage but does not show that high E-cadherin expression in colon epithelium is a limiting factor.

In humans colorectal cancer might develop over years not days and it cannot be excluded that tumours can develop from mutation of a single allele of b-catenin after a longer time period. As APC mediated cancer formation requires two genetic hits the advantage of APC over single hit b-catenin mutations becomes even less clear.

2. A second critical aspect of discussion is the interpretation of the finding that b-catenin starts to accumulate after 25 days leading to CPC in the small intestine (Suppl. Fig. 3). While the accumulation is obvious it is clear from a cell biological point of view that this cannot be simply due to lack of b-catenin degradation as a consequence of mutations. On a cellular level the maximal level of b-catenin should be reached within a few hours of mutation, determined by transcriptional and translational kinetics, and probably by mechanisms leading to baseline protein degradation, which has to be operative otherwise b-catenin would accumulate indefinitely.

In that respect it would be important to show biochemical quantifications of (cytosolic) b-catenin levels in the different genetic models to see whether indeed the levels vary according to the assumption by the authors.

3. Finally, the results obtained with combined GSK3 mutations are novel and interesting but do not really contribute to answering the main question of the paper.

In summary, the paper contains a lot of interesting mouse genetics and will be frequently quoted. The authors should tone down their argument that it explains "why APC and not b-catenin mutations are common in colorectal cancer" and should discuss the limitations of their models for human colorectal tumorigenesis.

Minor points. The heading "Switching GSK3 into a tumor suppressor gene in the intestine" (p. 8) is misleading.

What is meant by "aged cohorts" in Fig. 4? What is the precise age?

Referee #3:

Manuscript Title: "Precise Wnt signalling output explain why Apc and not β -catenin mutations are common in Colon Cancer"

In this study by Huels et al., gene knockouts in the mouse intestine are used to probe the basis behind the longstanding observation that APC mutations are far more abundant in colorectal cancer than stabilizing mutations in beta-catenin. Since high levels of unregulated Wnt signaling are the known cause of tumorigenesis in the intestine, it is a matter of debate why these two classes of mutations (both of which interfere with beta-catenin degradation in cells) should be so skewed in their occurrence in colon cancer. Using multiple Cre drivers to introduce gene mutations or gene knockouts, strategies for introducing inactivating or stabilizing mutations in beta-catenin, APC, or the GSK3alpha/beta kinase family as well as in vitro organoid culture assays, the authors show that Cre-driven generation of APC truncation mutations is more potent than beta-catenin stabilizing mutations in generating dysplasia and tumor burden. The authors also show that the potency of the beta-catenin mutations can be enhanced by heterozygosity at the E-cadherin locus. Overall the authors use their considerable expertise with genetic engineering in mice to support and further the idea that a "just-right" amount of beta-catenin is necessary for colon epithelial cell transformation. This is not a new idea and therefore probably the most unique new advancement in concept in the study here is a highlight of a possible correlation with the expression levels of E-cadherin (which are considerably higher in the mouse colon) - meaning, higher levels of Ecadherin expression raise the threshold needed for beta-catenin accumulation to be transformative. While this is interesting, it remains that there are outstanding unknowns about the model and the mechanism behind the tissue specificity of colon cancer - some of which should be considered in the author's study.

1. Previous studies (e.g. Leedham et al.) have shown a gradient of strong-to-weak Wnt signaling throughout the mouse and human intestine (strongest in the small intestine and weakest in the colon). The Leedham study highlighted the power of comparing mouse vs. human as it highlighted key similarities and differences between the two systems, and their reported findings are corroborated here by Huels et al. using mouse genetics. However, probably the most important disconnect to point out is that if it were simply levels of beta-catenin accumulation that were needed to transform and initiate tumorigenesis, one would predict a higher incidence of tumors in the human small intestine, similar to mice. This is not the case. The highest tumor burden in the human population is in the colon in a proximal to distal gradient. I am not sure that the authors can claim the straightforward conclusion as stated in the Abstract that it is simply a beta-catenin threshold that must be breached for tumor initiation.

2. The intestinal organoid system used is very powerful and it is interesting how the authors show that intestinal organoids from APC^{fl/fl} mice are independent of R-spondin. There are several issues with this data however. It is never stated in the legend or methods whether these organoids are from the small intestine or the colon. In the absence of that statement one presumes that they are from the small intestine as these are the easiest organoids to grow. There are published methods for the culture of colon organoids, and it would have enhanced the manuscript to i) compare Wnt dependence on colon organoids from the multiple mouse models, and ii) to quantitate the R-spondin independence (number of spheres, size of spheres, etc) - as the only quantitation provided for the organoid data is in Figure 6C,D. Organoids are cultured outside of their microenvironment of negative Wnt regulators (sFRPs, BMPs, etc.), and therefore they are more primed for Wnt signaling than normal - some measure of quantitation and comparison (small dose response curve to R-spondin?) would go further to substantiate their point.

3. When discussing the implications of the various mouse models, the manuscript focuses solely on beta-catenin availability. But of course APC plays many roles in cells, including nuclear import and occupancy of Wnt Response Elements on target genes where it facilitates removal of beta-catenin from sites. Similarly, GSK3beta plays multiple roles in Wnt signaling, including a positive one at the plasma membrane, and possibly in the nucleus as well. Finally, E-cadherin releases not only beta-catenin, but alpha-catenin which has been reported to traffic to the nucleus and assemble on

Wnt Response Elements. Therefore, each mouse model may not only influence total cellular beta-catenin, but the activities of the Frizzled:Lrp receptor complex on the membrane or the transcription regulatory complex on Wnt Response Elements in the nucleus. These influences may be wholly missing in the beta-catenin stabilizing mutations - and could account for the weaker phenotype.

4. The E-cadherin western blot experiment is incomplete. The authors should provide western blots showing total beta-catenin and "free beta-catenin" in both wildtype and mouse models where E-cadherin is heterozygous (not just the mRNA - which does not work as proxy for protein in the case of beta-catenin). If there are higher levels of E-cadherin protein in the colon, one presumes that there are also higher levels of beta-catenin protein to account for the greater number of adhesion complexes. The mouse model of E-cadherin heterozygosity is intriguing, but the limited organoid data (small intestine only), makes the conclusion tenuous and preliminary. More data on the relative levels of E-cadherin and beta-catenin in the various mouse models is warranted.

Resubmission

08 April 2015

Referee #1:

Review: Precise Wnt signaling output explain why Apc and not b-catenin mutations are common in Colon Cancer.

This manuscript aims to dissect the differential role of APC and b-catenin mutations in CRC (colorectal cancer). The authors present evidence that b-catenin mutations are rare in CRC and bi-allelic b-catenin activating mutations are required to drive Wnt transformation in the colon. They further suggest that a single activating mutation of b-catenin is not able to drive transformation due to high colonic levels of the b-catenin binding protein E-cadherin. Finally, the authors suggest that co-deletion of a single copy of E-cadherin in the colon synergizes with a mono-allelic b-catenin mutation to transform intestinal progenitor cells toward colon cancer. Unfortunately, the experimental design contains a number of flaws, and the conclusions drawn are not supported by the results as indicated as follows:

1. Data presented in Fig. 1 includes TCGA analysis, which the authors claim indicates that b-catenin mutations in CRC do not occur in exon 3. However, this exon contains the sites at which mutations have been shown to block phosphorylation and ubiquitination in CRC.

We apologize to the reviewer for not explaining clearly in the text. Our point was the fact that there are very very few mutations in exon 3 in sporadic colorectal cancer **not that they never occur**. The TCGA data, and recent publications (Luchtenborg, Weijenberg et al. 2005; Vasovcak, Pavlikova et al. 2011), indicate that CTNNB1 are rare in sporadic CRC (<5%). More importantly, most of these CTNNB1 mutations occur outside of exon 3 and are mostly unique in the whole sequencing portal, suggesting these mutations as passenger mutations. Indeed out of the TCGA sequencing data there are only 2/224 patients with exon 3 mutations (see supplementary figure 8). We hope this is explained well in the text. We've altered the title removing the emphasis on human CRC.

2. The authors fail to compare b-catenin levels target gene activation in APC vs b-catenin mutant tumors, which could help in testing their model.

We now present qRT-PCR results for these mouse models and this has been added to Supplementary Fig. 7.

3. A major weakness of the study is the complete lack of evidence that a single activating mutation of b-catenin is unable to drive Wnt transcriptional activity due to higher E-cadherin levels in the colon as compared to the small intestine. The authors do not provide any data supporting the proposed mechanism by which E-cadherin would sequester mutated b-catenin in the cytoplasm. The authors need to perform experiments to support this hypothesis as well as exclude different mechanisms involved in the lack of Wnt transcriptional activation by a single mutated b-catenin in the colon.

We thank the reviewer for this important point. We have now specifically analysed the E-cadherin:β-catenin complexes by immunoprecipitation and additionally quantified E-cadherin:β-catenin complexes at the membrane via Proximity ligation assay.

Importantly we show that the mutated form of β-catenin binds to E-cadherin. As the mutant form is smaller than wild type β-catenin this allowed us to do experiments to investigate the kinetics of mutant β-catenin accumulation with E-cadherin. We found in mice that are homozygous for exon 3 β-catenin (VilCre^{ER} Catnb^{lox(ex3)/lox(ex3)}) that the E-cadherin in the small intestine is rapidly saturated with mutant β-catenin (with nearly 90% of E-cadherin:β-catenin complexes being the mutant form of β-catenin). However this process takes longer in the colon (day 1 35% colon vs 71% Small intestine, day 2: 63% vs 86%). This fits with the data that there are more E-cadherin: β-catenin complexes in the colon than in the small intestine (Figure 3). In mice carrying a single mutant copy of β-catenin, the accumulation takes much longer than in a mouse homozygous for β-catenin. The genetic evidence shows that we can increase nuclear β-catenin by reducing E-cadherin expression (and complexes of E-Cadherin: β-catenin by ~50%, Supplementary Figure 6). This suggests that E-cadherin retention of β-catenin is limiting the transformation of the intestine by β-catenin exon 3 mutation.

This evidence is particularly critical, since b-catenin mutations in HCC or in pseudopapillary pancreatic carcinomas, which are mostly heterozygous, are sufficient to drive Wnt transcriptional activation and to participate in transformation as cited by the authors.

This is again an important point. We were able to obtain a tissue microarray array of patients with solid pseudopapillary tumours (SPT). This tumour is characterised by activating β-catenin mutations in exon 3 (110/122 of SPT have β-catenin mutations and these are all in exon 3, see supplementary figure 8). Importantly when we compare tumour to normal tissue there was a dramatic fall in E-cadherin:β-catenin complexes by PLA (Figure 6). As the reviewer suggested we also investigated expression of Wnt targets genes and E-cadherin from expression data of HCC (Ahn, Jang et al. 2014). We found a strong negative correlation that was significant between E-cadherin and several Wnt signalling targets. Unfortunately, we were unable to obtain tumour material that we knew contained exon 3 mutation so given the relatively low percentage of β-catenin mutations we focussed on this transcriptome analysis.

The authors could employ mass-spec studies to identify differential b-catenin partners in CRC, HCC, and pseudopapillary pancreatic carcinomas. The authors could also apply the same comparison using the engineered CRC mouse models carrying +/+, lox(ex3)/+, and lox(ex3)/lox(ex3) b-catenin.

We agree that this would be an interesting experiment. However given the results from the IPs of β-catenin to E-cadherin we believe that this explains the results we see. Also we did not have frozen material available for SPT, HCC or CRC with β-catenin exon 3 mutations. Given the clear fall of E-cadherin- β-catenin complex (associated with accumulation of β-catenin) in SPT, we believe we would mainly pull out nuclear factors that bind β-catenin e.g. TCF/LEFS so it is possible that this might dominate binding partner analysis when compared to other cancers.

4. In figure 5, E-cadherin levels should be measured in other tissues where b-catenin is mutated such as in HCC and pseudopapillary pancreatic carcinomas to further support its role in regulating b-catenin localization.

This was a very good suggestion and the data for this is in figure 6. The fall in E-cadherin:β-catenin complexes fits our hypothesis very well.

5. IP studies would be needed to establish whether higher E-cadherin levels are able to sequester mutated b-catenin in the cytoplasm.

This was a very good suggestion and the data is in figure 4. The data clearly showed that E-cadherin can sequester mutated β-catenin.

6. To assess Wnt transcriptional activation, the authors only tested (by IHC) a single target gene, which varies within different experiments presented (i.e. SOX9 in Figure 4 and c-MYC in Figure 7).

The authors need to evaluate more than one Wnt target in each set of experiments and consistently test these same genes in all experiments.

We have now provided extra Wnt target genes by QRT-PCR. We would also highlight that the nuclear accumulation of β -catenin, the CPC phenotype, the increased proliferation and the associated accumulation of Wnt targets provides strong evidence for Wnt activation.

7. The title is not supported by the results provided.

We have now modified the title.

8. The authors need to test their conclusions from mouse studies by evaluating the Wnt transcriptional gene signature on laser captured human CRC samples harboring either APC or CTNNB1 mutations by qPCR or RNAseq analysis.

As we mentioned earlier from sequencing studies only 2/224 human sporadic colon cancers had β -catenin exon 3 mutation which makes the ability to obtain this signature very different. Moreover we would argue that if these tumours are initiated by β -catenin they would need a further mutation to deregulate Wnt signalling and then they would adopt a Wnt programme that would look like Apc deficient colon cancer.

Referee #2:

The group has a longstanding experience in establishing and analysing Wnt dependent mouse models of intestinal tumorigenesis. Thus, from a technical point of view everything is done well and there is not much to criticise. Also, the results as such are clear-cut and well documented. However, it is the significance and interpretation of results which is a matter of criticism.

I In particular conclusions drawn in the manuscript title that these studies explain why b-catenin mutations are rare in human colorectal cancer are too simple. Results in the small intestine show that APC deleted mice or homozygously mutated b-catenin mice develop lesions termed CPC already after 5 days whereas heterozygous b-catenin mice do so after 25 days. While these results suggest that it is the dose of b-catenin that determines whether transformation becomes visible they do also show that a single b-catenin mutation in principle suffices, it just takes a bit longer. Revealingly, when expressed in epithelial stem cells using Lgr5Cre there was no difference in CPC development between the three genotypes showing that a single b-catenin is sufficient in the specific cellular background. By extension the finding that the heterozygous b-catenin mice did not show phenotypes after 25 days in the colon could be due to the limited period of observation. The finding that mutation of E-cadherin leads to CPC in b-catenin heterozygous b-catenin mice just underscores the relevance of b-catenin dosage but does not show that high E-cadherin expression in colon epithelium is a limiting factor.

We thank the reviewer for the comments. We have now reworded the text to focus on β -catenin dosages. We would argue that our genetic data where we delete E-cadherin and see a clear phenotype would suggest that this is limiting **in the mouse**.

In humans colorectal cancer might develop over years not days and it cannot be excluded that tumours can develop from mutation of a single allele of b-catenin after a longer time period. As APC mediated cancer formation requires two genetic hits the advantage of APC over single hit b-catenin mutations becomes even less clear.

The reviewers makes a good point however we would point out that the rates of β -catenin exon3 mutations in human CRC are very low (2/224 from the TCGA sequencing analysis). We have added extra discussion to address this point. If we were to extrapolate the work of the Winton group it would suggest that unless the β -catenin exon 3 mutations exerted an immediate phenotype, there is a strong probability that these mutations may be lost through neutral drift over a 20 day period.

2. A second critical aspect of discussion is the interpretation of the finding that b-catenin starts to accumulate after 25 days leading to CPC in the small intestine (Suppl. Fig. 3). While the accumulation is obvious it is clear from a cell biological point of view that this cannot be simply due to lack of b-catenin degradation as a consequence of mutations. On a cellular level the maximal level of b-catenin should be reached within a few hours of mutation, determined by transcriptional and translational kinetics, and probably by mechanisms leading to baseline protein degradation, which has to be operative otherwise b-catenin would accumulate indefinitely. In that respect it would be important to show biochemical quantifications of (cytosolic) b-catenin levels in the different genetic models to see whether indeed the levels vary according to the assumption by the authors.

The reviewer makes an important point so we have now used IP studies to examine E-cadherin:β-catenin complexes. Using this approach, we can show the kinetics of mutant β-catenin accumulation with E-cadherin (Figure 4) and its total accumulation. Importantly we show that the mutated form of β-catenin binds to E-cadherin. As the mutant form is smaller than wild type β-catenin this allowed us to do experiments to investigate the kinetics of mutant β-catenin accumulation with E-cadherin. We found in mice that are homozygous for exon 3 β-catenin (VilCre^{ER} Catnb^{lox(ex3)/lox(ex3)}) that the E-cadherin in the small intestine is rapidly saturated with mutant β-catenin (with nearly 90% of E-cadherin:β-catenin complexes being the mutant form of β-catenin). However this process takes longer in the colon (day 1: 35% colon vs 71% Small intestine, day 2: 63% vs 86%). This fits with the data that there are more E-cadherin: β-catenin complexes in the colon than in the small intestine (Figure 3). In mice carrying a single mutant copy of β-catenin, the accumulation takes much longer than in a mouse homozygous for β-catenin. The genetic evidence shows that we can increase nuclear β-catenin by reducing E-cadherin expression (and complexes of E-Cadherin: β-catenin by ~50%, Supplementary Figure 6). This suggests that E-cadherin retention of β-catenin is limiting the transformation of the intestine by β-catenin exon 3 mutation.

3. Finally, the results obtained with combined GSK3 mutations are novel and interesting but do not really contribute to answering the main question of the paper.

We have reorganised the manuscript considerably so that we now only use the GSK3 work to show the impact of when we lose the destruction complex by another means. This is now included only as a small part of Figure 1A and added the rest of the data to supplementary data (Suppl. Figure 1). We would be happy to remove this part completely, if requested by the reviewers but currently feel that the comparison of complete GSK3 loss to Apc loss is useful.

In summary, the paper contains a lot of interesting mouse genetics and will be frequently quoted. The authors should tone down their argument that it explains "why APC and not b-catenin mutations are common in colorectal cancer" and should discuss the limitations of their models for human colorectal tumorigenesis.

We have retitled and reworded the manuscript according to the reviewers comments. We have added a number of caveats and issues in the discussion.

Minor points. The heading "Switching GSK3 into a tumor suppressor gene in the intestine" (p. 8) is misleading.

We have now reworded this section.

What is meant by "aged cohorts" in Fig. 4? What is the precise age?

Ageing cohorts meant mice were aged until they developed signs of intestinal neoplasia (weight loss, paling feet, hunching).

Referee #3:

Manuscript Title: "Precise Wnt signalling output explain why Apc and not β -catenin mutations are common in Colon Cancer"

In this study by Huels et al., gene knockouts in the mouse intestine are used to probe the basis behind the longstanding observation that APC mutations are far more abundant in colorectal cancer than stabilizing mutations in beta-catenin. Since high levels of unregulated Wnt signaling are the known cause of tumorigenesis in the intestine, it is a matter of debate why these two classes of mutations (both of which interfere with beta-catenin degradation in cells) should be so skewed in their occurrence in colon cancer. Using multiple Cre drivers to introduce gene mutations or gene knockouts, strategies for introducing inactivating or stabilizing mutations in beta-catenin, APC, or the GSK3alpha/beta kinase family as well as in vitro organoid culture assays, the authors show that Cre-driven generation of APC truncation mutations is more potent than beta-catenin stabilizing mutations in generating dysplasia and tumor burden. The authors also show that the potency of the beta-catenin mutations can be enhanced by heterozygosity at the E-cadherin locus. Overall the authors use their considerable expertise with genetic engineering in mice to support and further the idea that a "just-right" amount of beta-catenin is necessary for colon epithelial cell transformation. This is not a new idea and therefore probably the most unique new advancement in concept in the study here is a highlight of a possible correlation with the expression levels of E-cadherin (which are considerably higher in the mouse colon) - meaning, higher levels of Ecadherin expression raise the threshold needed for beta-catenin accumulation to be transformative. While this is interesting, it remains that there are outstanding unknowns about the model and the mechanism behind the tissue specificity of colon cancer - some of which should be considered in the author's study.

1. Previous studies (e.g. Leedham et al.) have shown a gradient of strong-to-weak Wnt signaling throughout the mouse and human intestine (strongest in the small intestine and weakest in the colon). The Leedham study highlighted the power of comparing mouse vs. human as it highlighted key similarities and differences between the two systems, and their reported findings are corroborated here by Huels et al. using mouse genetics. However, probably the most important disconnect to point out is that if it were simply levels of beta-catenin accumulation that were needed to transform and initiate tumorigenesis, one would predict a higher incidence of tumors in the human small intestine, similar to mice. This is not the case. The highest tumor burden in the human population is in the colon in a proximal to distal gradient. I am not sure that the authors can claim the straightforward conclusion as stated in the Abstract that it is simply a beta-catenin threshold that must be breached for tumor initiation.

We have adjusted the interpretation and translation of our findings to human cancer. The limiting effect of E-cadherin levels on CTNNB1 mutations was analysed in more human cancers specifically SPT which is characterised by a reduction of E-cadherin: β -catenin complexes. We have also completely reworked the discussion to cover the reviewer concerns.

2. The intestinal organoid system used is very powerful and it is interesting how the authors show that intestinal organoids from APC^{fl/fl} mice are independent of R-spondin. There are several issues with this data however. It is never stated in the legend or methods whether these organoids are from the small intestine or the colon. In the absence of that statement one presumes that they are from the small intestine as these are the easiest organoids to grow. There are published methods for the culture of colon organoids, and it would have enhanced the manuscript to i) compare Wnt dependence on colon organoids from the multiple mouse models,

As correctly assumed by the reviewer, we focused on the small intestinal organoid culture system and it's comparison between the different genotypes. Unfortunately the higher EDTA extraction protocols required to remove colonic crypts affected the E-cadherin: β -catenin ratio. We also found the ratio of E-cadherin: β -catenin between the small

intestine and the colon is profoundly altered in culture.

Figure 1 PLA for E-cadherin:β-catenin in purified crypts. Crypts from the small intestine (SI) and the colon (col) of wildtype mice were purified after 30min incubation in EDTA(2mM and 25mM, respectively). The EDTA step already diminishes the increased E-cadherin:β-catenin levels observed *in vivo*.

Figure 2 VilCreER APC^{-/-} crypts grown in crypt culture conditions change the number of E-cadherin:β-catenin complexes from *in vivo* to *in vitro*. **A)** PLA quantification of crypts from VilCre APC^{fl/fl} mice, small intestine and colon, show a similar pattern as observed in wildtype mice with more complexes in the colon. **B)** Spheres from VilCre APC^{fl/fl} mice after 2 weeks in culture have more E-cadherin:β-catenin complexes when they originate from the small intestine, than from the colon. N=3, 10 spheres per mouse were analysed for the mean. Error bars s.e.m.

Figure 3 AhCre^{ER} Catnb^{lox(ex3)/+} (BCAT ex3/+) and AhCre^{ER} Catnb^{lox(ex3)/+} Cdh1^{fl/+} (Bcat ex3/+ Ecad fl/+) culture. 300 crypts/well from the small intestine and the colon were plated in 3 different experiments (N=3). In contrast to crypts from the small intestine, colonic crypts were able to survive in culture. Error bars s.e.m.

We would be more than happy to add these results to the supplementary data to explain why we did not use the colon spheres more.

and ii) to quantitate the R-spondin independence (number of spheres, size of spheres, etc) - as the only quantitation provided for the organoid data is in Figure 6C,D. Organoids are cultured outside of their microenvironment of negative Wnt regulators (sFRPs, BMPs, etc.), and therefore they are more primed for Wnt signaling than normal - some measure of quantitation and comparison (small dose response curve to R-spondin?) would go further to substantiate their point.

We have now added qualitative quantification of the organoid shape (organoid-like vs. sphere) in Figure 1E.

3. When discussing the implications of the various mouse models, the manuscript focuses solely on beta-catenin availability. But of course APC plays many roles in cells, including nuclear import and occupancy of Wnt Response Elements on target genes where it facilitates removal of beta-catenin from sites. Similarly, GSK3beta plays multiple roles in Wnt signaling, including a positive one at the plasma membrane, and possibly in the nucleus as well. Finally, E-cadherin releases not only beta-catenin, but alpha-catenin which has been reported to traffic to the nucleus and assemble on Wnt Response Elements. Therefore, each mouse model may not only influence total cellular beta-catenin, but the activities of the Frizzled:Lrp receptor complex on the membrane or the transcription regulatory complex on Wnt Response Elements in the nucleus. These influences may be wholly missing in the beta-catenin stabilizing mutations - and could account for the weaker phenotype.

We agree with the reviewer that there are many potential other effects of both APC and GSK3 deletion that might alter Wnt signalling. However, we believe the comparison between one and two copies of b-catenin exon mutation is crucial. The finding that 2 copies of b-catenin exon 3 produces a similar phenotype e.g. nuclear b-catenin, Wnt target gene activation (*Myc* and *Axin2*) and the CPC phenotype shows that preventing the destruction of all b-catenin recapitulates either *Apc* and GSK3 loss. This suggests that in the normal situation that E-cadherin binds b-catenin and on release, free b-catenin is turned over by the destruction complex. When we have mutant b-catenin this still binds to E-cadherin and thus when there is only a single copy this takes longer to saturate E-cadherin. Therefore there is less free b-catenin (which cannot be destroyed) and hence less accumulation in the nucleus. Overall we believe that the explanation for our phenotypes is the accumulation of b-catenin rather than any other impacts of GSK3 or APC on Wnt signalling.

4. The E-cadherin western blot experiment is incomplete. The authors should provide western blots

showing total beta-catenin and "free beta-catenin" in both wildtype and mouse models where E-cadherin is heterozygous (not just the mRNA - which does not work as proxy for protein in the case of beta-catenin). If there are higher levels of E-cadherin protein in the colon, one presumes that there are also higher levels of beta-catenin protein to account for the greater number of adhesion complexes.

To examine this further, we have performed the proximity ligation assays to quantify the number of E-cadherin: b-catenin complexes in these mouse models (Figure 3). Importantly we see a very good correlation between E-cadherin expression by qRT-PCR, by western analysis and by the PLA (Figure 3). Furthermore genetic reduction of E-cadherin (AhCre^{ER} Cdh1^{fl/+}) reduced E-cadherin expression by almost 50% with a similar reduction in complexes by PLA (Supplementary figure 6). Therefore, in this case the E-cadherin expression can be used as a proxy for the protein levels. The reviewer was right that there is more b-catenin protein in the colon epithelium than the small intestine (as seen *in situ* with the PLA) (Figure 3).

The mouse model of E-cadherin heterozygosity is intriguing, but the limited organoid data (small intestine only), makes the conclusion tenuous and preliminary. More data on the relative levels of E-cadherin and beta-catenin in the various mouse models is warranted.

We have added much more data now on the relative levels of E-cadherin-b-catenin in the intestinal tissue, particularly the incorporation of the PLA work in our mouse models and the E-cadherin-b-catenin IP work.

Importantly we show that the mutated form of β -catenin binds to E-cadherin. As the mutant form is smaller than wild type β -catenin this allowed us to do experiments to investigate the kinetics of mutant β -catenin accumulation with E-cadherin. We found in mice that are homozygous for exon 3 β -catenin (VilCre^{ER} Catnb^{lox(ex3)/lox(ex3)}) that the E-cadherin in the small intestine is rapidly saturated with mutant β -catenin (with nearly 90% of E-cadherin:b-catenin complexes being the mutant form of β -catenin). However this process takes longer in the colon (day 1: 35% colon vs 71% Small intestine, day 2: 63% vs 86%). This fits with the data that there are more E-cadherin: β -catenin complexes in the colon than in the small intestine (Figure 3). In mice carrying a single mutant copy of β -catenin, the accumulation takes much longer than in a mouse homozygous for β -catenin. The genetic evidence shows that we can increase nuclear β -catenin by reducing E-cadherin expression (and complexes of E-Cadherin: β -catenin by ~50%, Supplementary Figure 6). This suggests that E-cadherin retention of β -catenin is limiting the transformation of the intestine by β -catenin exon 3 mutation.

Given the alterations in the PLA when we put the colonic organoids in culture (EDTA is known to disrupt E-cadherin), we further increased the *in vivo* validation in the colon by intercrossing the Catnb^{lox(ex3)/+} CDH1^{fl/+} mice to Lgr5Cre^{ER} and we now see the development of colonic lesions (in line with the AhCre^{ER} and VilCre^{ER} data).

References.

- Ahn, S. M., S. J. Jang, et al. (2014). "Genomic portrait of resectable hepatocellular carcinomas: implications of RB1 and FGF19 aberrations for patient stratification." *Hepatology* **60**(6): 1972-1982.
- Luchtenborg, M., M. P. Weijnen, et al. (2005). "Mutations in APC, CTNNB1 and K-ras genes and expression of hMLH1 in sporadic colorectal carcinomas from the Netherlands Cohort Study." *BMC Cancer* **5**: 160.
- Vasovcak, P., K. Pavlikova, et al. (2011). "Molecular genetic analysis of 103 sporadic colorectal tumours in Czech patients." *PLoS One* **6**(8): e24114.

Thank you for the submission of your revised manuscript to The EMBO Journal. As you will see below, your article was sent to former referee #3, who now considers that you have properly dealt with the main concerns originally raised in the review process, and therefore I am writing with an 'accept in principle' decision, which means that I will be happy to formally accept your manuscript for publication once a few more minor issues have been addressed.

As I said, referee #3 now believes that all major concerns have been addressed and your manuscript is almost ready for publication (see below). Only minor issues regarding some clarifications and statistics remain. As a guide, statistical analyses must be described either in the Materials and Methods section or in the legend of the figure to which they apply, and will include a definition of the error bars used and the number of independent experiments performed. Significance analyses must also be specified if used.

Browsing through the manuscript myself I have also noticed a few small issues with data presentation. Some micrographs throughout the manuscript lack scale bars, which we require for clarity.

As you might know, every article now includes a 'Synopsis' to further enhance their discoverability. Synopses are displayed on the html version of the article and they are freely accessible to all readers. The synopsis includes an image, as well as 2-5 one-short-sentence bullet points that summarize the article. I would be grateful if you could provide both the figure and the bullet points for your article.

If you have any questions or need any further input, please do not hesitate to contact me.

Thank you very much for your patience. I am looking forward to seeing the final version of your manuscript. Congratulations in advance for a successful publication.

 REFEREE COMMENTS

Referee #3:

Manuscript Title: "E-cadherin can limit the transforming properties of activating beta-catenin mutations"

This revised study by Huels et al., shows that the abundance of E-cadherin complexes in the mouse colon might act as a tumor-limiting buffer for stabilized pools of beta-catenin that derive from heterozygous mutation of exon3 of the CTNNB1 locus. This revision clarifies many of the reviewer's questions, concerns and importantly provides new data that directly quantitate E-cadherin:beta-catenin complexes (use of the powerful Proximity Ligation Assay; Fig. 3D and S6). The new data shows that there are more E-cadherin:beta-catenin complexes in the colon and that E-cadherin pools in the small intestine quickly become saturated with stabilized, mutant beta-catenin. The data also show rather conclusively that reduction of E-cadherin levels increases the "potency" of beta-catenin exon3 mutations and tumors (or dysplastic lesions) form more readily. Overall the data contribute to an understanding of the cell biology and complex dynamics with E-cadherin, and they raise the interesting point that different levels of E-cadherin in tissues might be a significant factor in influencing which types of Wnt pathway mutations are able to cause cellular transformation. Some specific points below:

1. The Leedham study is now cited and discussed in the Discussion section. This is an important inclusion in the manuscript as readers may or may not be aware of the gradient of Wnt signaling throughout the intestinal tract. The authors have also expanded their discussion to address the obvious question/conundrum of Wnt pathway mutations in colon cancer (with low incidence of tumors in the small intestine - a cancer phenotype directly opposed to the mouse phenotype).

2. Colon organoid data: The authors provide additional data in their Response to Reviewers. This is a welcome addition and this reviewer acknowledges the extra effort on the part of the authors, but the new data is unfortunately confusing and it raises some questions. For example, while this reviewer agrees with the authors that colon organoids are difficult to maintain *in vitro* compared to small intestine organoids, the implication of their study is that this is due to low Wnt signaling (high E cadherin complexes). Yet in Figure 1 and 2 of the new data in the Response to reviewers, the authors show that there are actually equal levels of Ecadherin:beta-catenin complexes in wildtype crypts derived from small intestine and from colon. There are even fewer Ecadherin:beta-catenin complexes in *VilCreER APC^{-/-}* colon crypts compared to small intestine. If the author's model is that the main driver of crypt survival is lower levels of Ecadherin (and thus greater Wnt signaling), why don't the colon crypts survive better than the small intestinal crypts? Is R-spondin added to these cultures? This is not explained.

3. Figure 3 in the Response shows extremely low survival data for crypts with heterozygous beta-catenin. One presumes that this data is in the absence of R-spondin? If not, and R-Spondin is present, how is it that wildtype crypts shown in Figure 1 of the manuscript appear to survive much better? This reviewer is guessing that there is no R-spondin added to the cultures, but there is a paucity of text describing the experiment. The reviewers have offered to include these data as supplementary data in the manuscript. In general, I agree that they should include this data, but with careful and complete descriptions of the methods used, otherwise the graphs introduce confusion.

4. Statistics: Supplemental Figure 6 is a good addition to this revised manuscript, but Panel B showing Proximity Ligation Assay data is not explained very well in the legend. The mean and the S.E.M. are shown for each condition, but there are extra data points (dots) on each bar and they are not defined. There are no statistics either - this is especially important for the colon data. What is the p-value?

Revision - authors' response

23 June 2015

Thank you for your positive response upon our manuscript. We have now addressed all the remaining minor issues.

1. We have included all of the statistical tests that we have used and the sample sizes. Importantly we have replaced our bar graphs with univariate scatter plots according to the recent publication in Plos Biology "Beyond bar and line graphs".
2. We have now included 2 new supplementary figures with an improved description of the data that was previously included in the rebuttal letter.

Thanks again for your interest in our work. We have had a very good experience in working with your journal for this manuscript.